# Annexin A5 as an immune checkpoint inhibitor and tumor-homing molecule for cancer treatment

Tae Heung Kang[1,6], Jung Hwa Park[1,6], Andrew Yang [2,3], Hyun Jin Park[1], Sung Eun Lee[1], Young Seob Kim[1], Gun-Young Jang[1], Emily Farmer[3], Brandon Lam[4], Yeong-Min Park[1,7✉] & Chien-Fu Hung [5,7]

The interaction between immune cells and phosphatidylserine (PS) molecules exposed on the surface of apoptotic-tumor bodies, such as those induced by chemotherapies, contributes to the formation of an immunosuppressive tumor microenvironment (TME). Annexin A5 (AnxA5) binds with high affinity to PS externalized by apoptotic cells, thereby hindering their interaction with immune cells. Here, we show that AnxA5 administration rescue the immunosuppressive state of the TME induced by chemotherapy. Due to the preferential homing of AnxA5 to the TME enriched with PS+ tumor cells, we demonstrate in vivo that fusing tumor-antigen peptide to AnxA5 significantly enhances its immunogenicity and anti-tumor efficacy when administered after chemotherapy. Also, the therapeutic antitumor effect of an AnxA5-peptide fusion can be further enhanced by administration of other immune checkpoint inhibitors. Our findings support the administration of AnxA5 following chemotherapy as a promising immune checkpoint inhibitor for cancer treatment.

[1] Department of Immunology, KU Open Innovation Center, School of Medicine, Konkuk University, Chungju, South Korea. [2] Medical Scientist Training Program, Baylor College of Medicine, Houston, TX, USA. [3] Department of Pathology, Johns Hopkins Medical Institutions, Baltimore, MD, USA. [4] Graduate Program in Immunology, Department of Pathology, Johns Hopkins Medical Institutions, Baltimore, MD, USA. [5] Department of Pathology, Department of Oncology, Johns Hopkins Medical Institutions, Baltimore, MD, USA. [6] These authors contributed equally: Tae Heung Kang, Jung Hwa Park. [7] These authors jointly supervised this work: Yeong-Min Park, Chien-Fu Hung. ✉email: immun3023@kku.ac.kr; chung2@jhmi.edu

As cells undergo apoptosis, the phosphatidylserine (PS) that normally resides in the inner leaflet of the plasma membrane undergoes relocalization to the outer plasma membrane where it becomes exposed to the extracellular environment[1]. Once externalized, PS interacts with PS receptors that are expressed on the vast majority of phagocytes, which serve as an eat me signal and promote phagocytic uptake of apoptotic cells[2,3]. In addition, it is observed that effective efferocytosis of apoptotic cells are associated with subsequent suppression of pro-inflammatory cytokines as well as the production of anti-inflammatory cytokines (for review see ref. [4]). Together, these effects help maintain homeostasis of the healthy body and prevent the generation of undesirable inflammatory response during normal cell death by inducing immune suppressive clearance of dying, apoptotic cells[1].

Although the intended function of PS-mediated clearance of apoptotic cells is to prevent the development of autoimmune responses against self-antigens, the same pathways are hijacked by cancers to suppress the generation of antitumor immunity (for review see ref. [5]). It is well recognized that a variety of conditions in the tumor microenvironment (TME), such as hypoxia, the presence of oxygen radicals, or exposure to anti-cancer treatments including irradiation and chemotherapy, can cause cellular stress and promote the externalization of PS by stressed tumor cells and apoptotic-tumor bodies[5,6]. While we and others have previously demonstrated that many anti-cancer drugs, such as chemotherapy, possess immunostimulating effects and enhance the immunogenicity and efficacy of anti-cancer immunotherapy[7-10], chemotherapy-induced elevation of exposed-PS may in turn contribute to the formation of an immunosuppressive TME, thereby restricting the functions of tumor-targeting immune cells and preventing the immune clearance of tumors[11-13]. Due to the role of PS dysregulation in the formation of an immunosuppressive TME, strategies inhibiting PS signaling have been explored as potentially attractive methods to enhance the potency of antitumor immunity[14-16].

Annexin A5 (AnxA5) is a member of the calcium and phospholipid binding protein family called the annexins, which bind PS with high affinity[17]. Due to its preferential PS binding property, AnxA5 has been utilized as a marker for the detection of cells undergoing apoptosis[18,19]. In addition, it has been shown that the binding of AnxA5 to PS+ apoptotic cells can modulate the PS-mediated immunosuppressive clearance, thereby increasing the immunogenicity of apoptotic cells, including irradiated, apoptotic-tumor cells[20-23]. Furthermore, due to the contribution of PS to immunosuppressive TME, AnxA5 may serve as a potential immune checkpoint inhibitor to enhance the immunogenicity of tumor-antigen specific immunotherapies, particularly following the administration of cytotoxic chemotherapeutics[5,24].

Here we show that administration of AnxA5 alleviates the immunosuppressive properties of TME generated by chemotherapy and enhances the immunogenicity and antitumor efficacy of tumor antigen-specific immunization. The potency of AnxA5 for immune checkpoint blockade therapy is comparable to that of other reported immune checkpoint inhibitors, including anti-PD-1, anti-PD-L1, anti-TIM-3, and anti-TGF-β. Furthermore, we demonstrated that AnxA5 can also serve as a homing molecule to concentrate AnxA5-linked tumor-antigens into PS-rich TME to enhance the magnitude of localized antitumor immunity. Lastly, AnxA5 treatment can be combined with immune checkpoint inhibitors targeting other signaling pathways for the induction of elevated, synergistic antitumor immune response. Our data support the use of AnxA5 following chemotherapy as a promising immune checkpoint inhibitor for cancer treatment.

## Results

**Annexin A5 protein generates potent therapeutic effects.** To evaluate the influence of AnxA5 administration on the therapeutic antitumor efficacy and immunogenicity of tumor-specific immunization following cisplatin treatment, we utilized a well-established preclinical tumor model with defined tumor-specific antigen, the TC-1 tumor model. TC-1 tumor cells express the E7 early protein of human papillomavirus type 16 (HPV16-E7)[25]. The TC-1 preclinical tumor model has been widely used as a clinically relevant model of HPV16-associated cancers, as HPV16 is responsible for causing numerous cancers in human patients, including 50% of all cervical cancer cases and 30% of all head and neck cancer cases (for review see refs. [26,27]). TC-1 tumor cells were inoculated subcutaneously into C57BL/6 mice to establish preclinical tumors. Tumor-bearing mice were then treated with intraperitoneal injection of cisplatin, intratumoral vaccination of HPV16-E7 long peptides, and/or intravenous administration of AnxA5 protein, with phosphate buffer saline (PBS) as negative controls (Fig. 1a). Of note, we have previously demonstrated in this model that chemotherapy exerts adjuvant effect to enhance the immunogenicity of the TME, permitting the generation of antitumor immune responses by subsequent intratumoral administration of HPV16-E7 long peptide without the need of additional adjuvant[9]. Following tumor challenge, tumor growth was monitored by palpation and inspection twice a week. Concurrent administration of cisplatin, E7 long peptide, and AnxA5 resulted in potent control of TC-1 tumor growth compared to other treatments (Fig. 1b). Furthermore, all TC-1 tumor-bearing mice treated with cisplatin, E7 long peptide, and AnxA5 survived at least 70 days after tumor challenge, while 60% of mice in the cisplatin and E7 long peptide group died by day 50 after tumor challenge, and all mice in remaining treatment groups died by day 40 after tumor challenge (Fig. 1c). When assessing the resultant E7-specific immune response generated by the various treatment strategies, significantly stronger systemic and tumor-infiltrating E7-specific CD8+ T cell responses were detected in TC-1 tumor-bearing mice treated with cisplatin, E7 long peptide, and AnxA5, as compared to mice in other treatment groups (Fig. 1d, e). These data suggest that AnxA5 administration can enhance the antitumor immunogenicity of therapeutic cancer vaccines following cisplatin treatment.

**Annexin A5 treatment rescued the immune suppressive TME.** To decipher the mechanism of AnxA5 administration in enhancing the generation of antitumor immune responses following cisplatin treatment, we sought to characterize the influence of AnxA5 administration in the TC-1 TME following cisplatin treatment. Following TC-1 tumor challenge, cisplatin treatment, and/or AnxA5 administration, we harvested the tumor tissues from tumor-bearing mice and assessed the presence of various immune cell populations and cytokines within the TME (Fig. 2a). Compared to cisplatin treatment only, subsequent administration of AnxA5 did not increase the population of CD11b+F4/80+ macrophages in the TME. However, compared to no treatment control, cisplatin treatment skewed the macrophage population towards the immunosuppressive M2 phenotype, which was restored by subsequent AnxA5 treatment (Fig. 2b). AnxA5 administration following cisplatin treatment also led to greater infiltration of CD8+ T cells and CD4+ T cells into the TME, while reducing the populations of immunosuppressive regulatory T cells (Tregs) and myeloid-derived suppressive cells (MDSCs), as well as the expression of PD-L1 within the TME (Fig. 2c, d). The reduction in immunosuppressive MDSC and Treg populations was also observed in the spleens of tumor-bearing mice following cisplatin and AnxA5 treatment

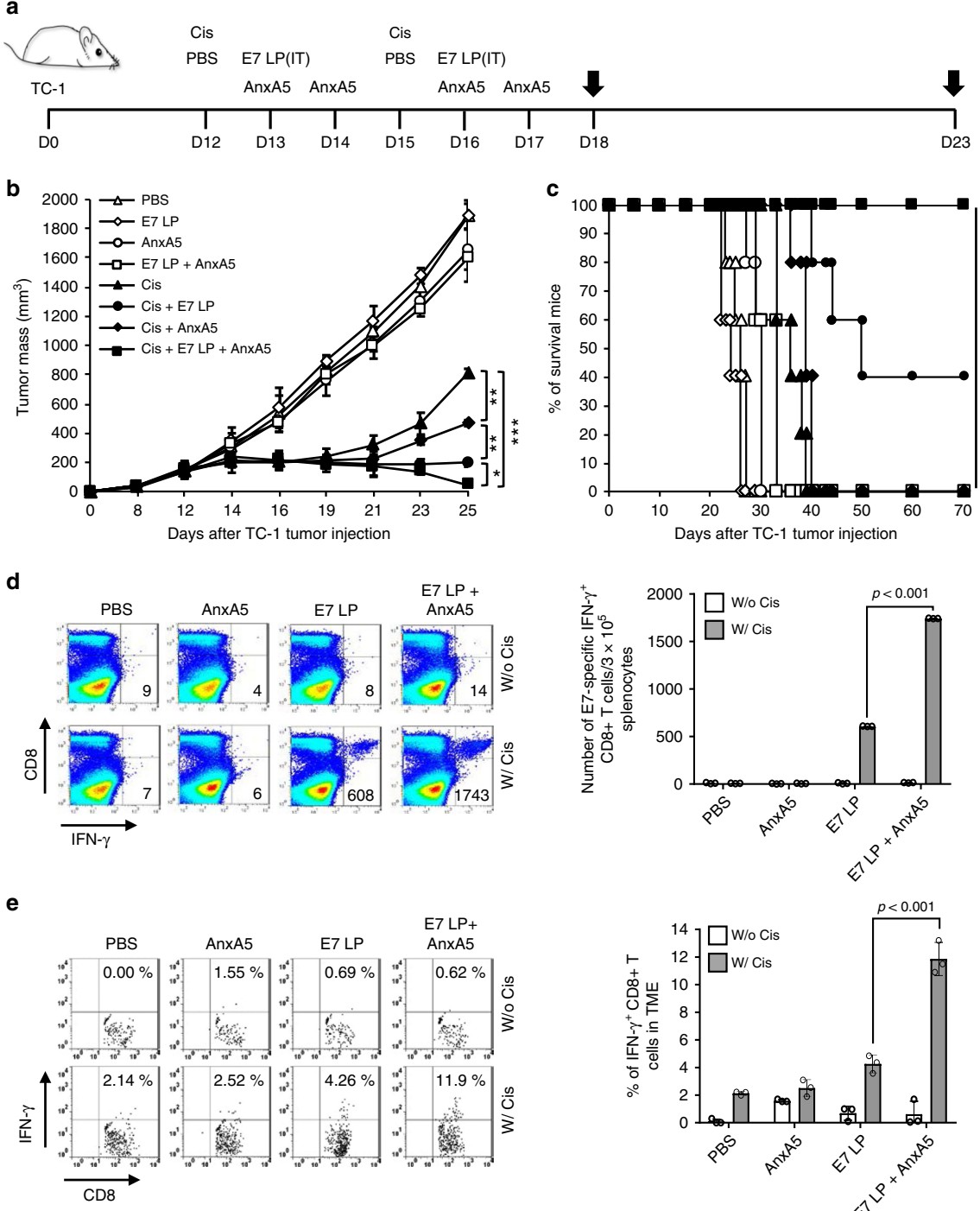

**Fig. 1 Therapeutic antitumor effect of Annexin A5 protein administration.** C57BL/6 mice (10 per group) were injected with $2 \times 10^5$ TC-1 cells/mouse subcutaneously on day 0. Mice were then treated intraperitoneally with 5 mg/kg Cisplatin on days 12 and 15, intravenously with 200jig/mice of Annexin A5 proteins on days 13, 14, 16, and 17, and/or intratumorally with 20jig/mice of E7 long peptide on days 13 and 16. The treatment groups are as follows: opened triangle - PBS only; opened sphere—E7 long peptide only; opened circle—Annexin A5 only; opened square—E7 long peptide and Annexin A5; closed triangle—cisplatin only; closed circle—cisplatin and E7 long peptide; closed sphere—cisplatin and Annexin A5; closed square—cisplatin, E7 long peptide, and Annexin A5. **a** Schematic diagram. **b** Line graph depicting TC-1 tumor growth in different treatment groups over time ($n = 10$). *P*-values were determined by one-way ANOVA and Turkey's test. **c** Kaplan–Meier survival analysis of TC-1 tumor-bearing mice in different treatment groups and the overall *P*-value was calculated by the log-rank test ($n = 10$). **d, e** On days 18 and 23, tumor tissues and spleens of TC-1 tumor-bearing mice in different treatment groups were harvested and analyzed for CD8+IFN-γ+ T cells by flow cytometry analysis, respectively. **d** Representative flow cytometry analysis and bar graph depicting the abundance of CD8+IFN-γ+ T cells in splenocytes of TC-1 tumor-bearing mice in different treatment groups. **e** Representative flow cytometry analysis and bar graph depicting the abundance of CD8+IFN-γ+tumor-infiltrating T cells in TC-1 tumor-bearing mice in different treatment groups. The error bars indicate mean ± SD. *P*-values were analyzed by Student's *t* test ($n = 3$). The results are representative of one of three independent experiments. Source data are provided as a Source Data file.

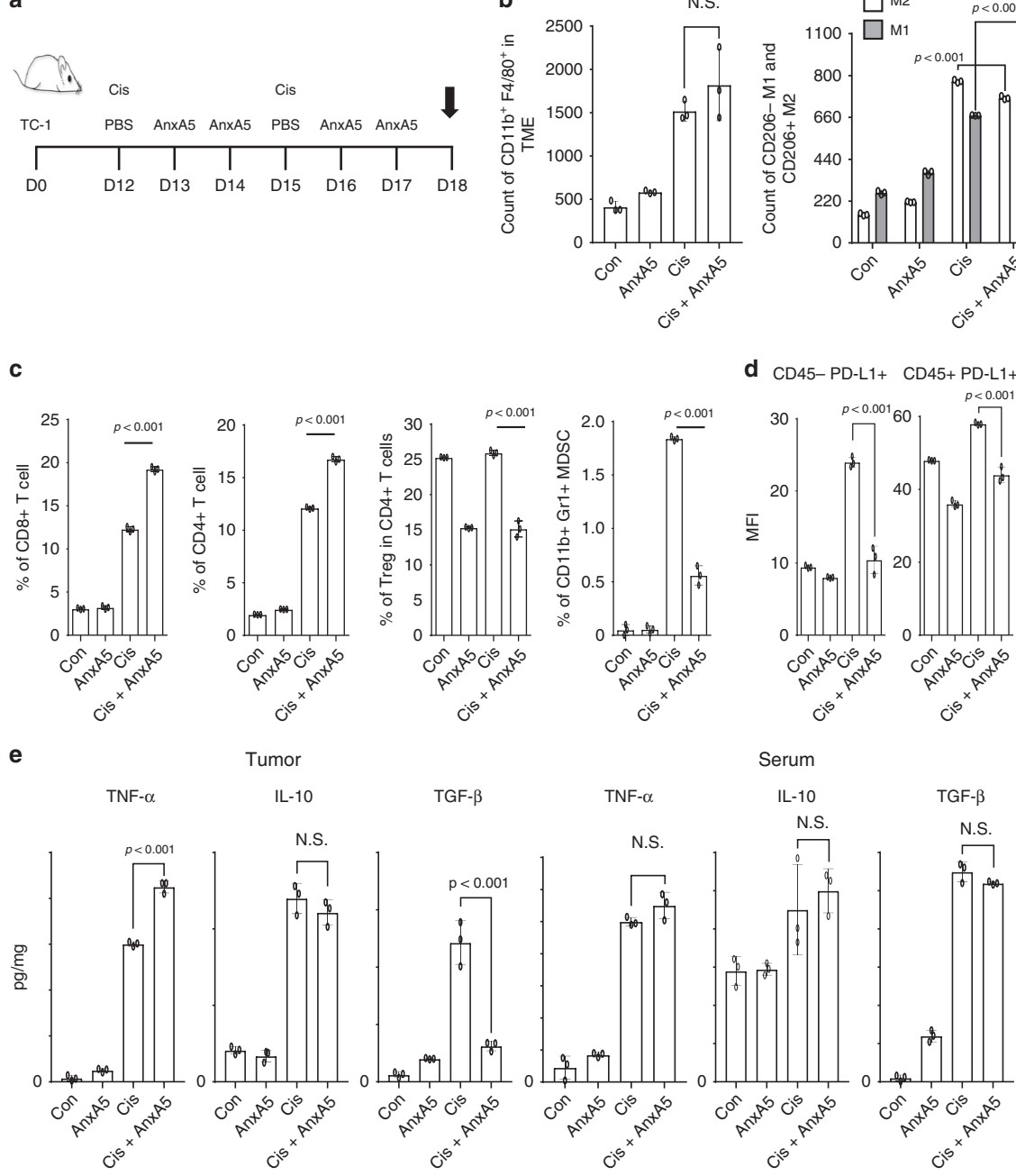

**Fig. 2 Characterization of tumor microenvironment following Annexin A5 treatment.** C57BL/6 mice (10 per group) were injected with $2 \times 10^5$ TC-1 cells/ mouse subcutaneously on day 0. Mice were then treated intraperitoneally with 5 mg/kg Cisplatin on days 12 and 15, and/or intravenously with 200jig/ mice of Annexin A5 proteins on days 13, 14, 16, and 17. PBS was used as control. On day 18, tumor tissues and serum of mice were harvested. **a** Schematic diagram. **b** Bar graphs depicting the abundance of CD11b+ F4/80+ macrophages and their M1/M2 distributions in the tumor tissue following flow cytometry analysis ($n = 3$). **c** Bar graphs depicting the presence of CD8+ T cells, CD4+ T cells, Treg cells, and MDSCs in the tumor tissue following flow cytometry analysis ($n = 3$). **d** Bar graphs depicting the expression of PD-L1 by CD45+ immune cells and CD45- tumor cells following flow cytometry analysis ($n = 3$). **e** Bar graphs depicting the levels of TNF-α, IL-10 and TGF-3 cytokines in the tumor tissue and serum of mice as measured by ELISA ($n = 3$). The error bars indicate mean ± SD. N.S. = not significant. For (**b-e**), $P$-values were analyzed by Student's $t$ test. The results are representative of one of three independent experiments. Source data are provided as a Source Data file.

(Supplementary Fig. 1). Furthermore, AnxA5 administration following cisplatin treatment enhanced the frequency of CD40hi and CD80hi dendritic cells (DCs) in the tumor draining lymph nodes, as well as the migration of DCs from the TME to these draining lymph nodes (Supplementary Fig. 2).

Since chemotherapy has been shown to enhance the externalization of PS by stressed and apoptotic-tumor cells[14] and the interaction between PS+ tumor cells with phagocytic innate

immune cells help promote the immunosuppressive state of TME (reviewed in ref. [5]), we hypothesize that AnxA5, by binding to PS, can prevent the induction of immune suppression by PS+ tumor cells. As shown in Supplementary Fig. 3, in vitro treatment with AnxA5 promotes the secretion of pro-inflammatory cytokines TNF-α and IL-12 and suppresses the production of anti-inflammatory, immunosuppressive cytokine TGF-β by bone-marrow-derived dendritic cells (BMDCs) and bone-marrow-

derived macrophages (BMDMs) co-cultured with cisplatin-treated apoptotic TC-1 tumor cells. Similarly, an increase in the TNF-α and IL-12 cytokine levels and a decrease in TGF-β cytokine level were observed in the TME of TC-1 tumor-bearing mice treated with cisplatin and AnxA5 as compared to those treated with cisplatin only (Fig. 2e). Cisplatin treatment also increased the level of MDSCs and Tregs attracting chemokine CCL2 in the TME[28,29], which was reduced by additional treatment with AnxA5 (Supplementary Fig. 4).

In addition to inducing the secretion of immunosuppressive cytokines by phagocytes, PS+ apoptotic-tumor cells can also exert inhibitory signals against tumor-specific CD8+ T cells via direct interaction (reviewed in ref. [30]). Indeed, in vitro incubation with soluble PS suppresses the activation and proliferation of CD8+ T cells stimulated with anti-CD3a antibody or PMA/I, which can be rescued by further treatment with AnxA5 (Supplementary Fig. 5a–d). Similarly, in vitro AnxA5 treatment enhances the activation and proliferation of OT-1 T cells incubated with ova-transfected TC-1 (TC-1 ova) cells that are pretreated with or without cisplatin (Supplementary Fig. 5e, f).

Together, these data support the notion that cisplatin treatment can contribute to an immunosuppressive TME by promoting the formation of PS-exposed apoptotic-tumor bodies. These data also suggest that AnxA5 administration helps to block the interaction of PS+ tumor cells with innate and adaptive immune cells, thereby rescuing the immunosuppressive state of TME in an antigen-independent manner.

**Contrasting therapeutic effects of TGF-β and TNF-α antibody.** Since we observed that AnxA5 administration following cisplatin treatment increased the level of TNF-a and decreased the level of TGF-3 within the TME, we sought to characterize how the alteration of these cytokine levels affects the TME by treating TC-1 tumor-bearing mice receiving cisplatin and/or E7 long peptide administration with anti-TNF-a or anti-TGF-3 blocking antibody (Fig. 3a). Compared to those treated with cisplatin and E7 long peptide only, significantly better tumor control and prolonged survival were observed in TC-1 tumor-bearing mice treated with cisplatin, E7 long peptide, and anti-TGF-3, while treatment with cisplatin, E7 long peptide, and anti-TNF-a resulted in worse tumor control and mouse survival (Fig. 3b, c). Administration of anti-TGF-3 also enhanced both the systemic and tumor-infiltrating E7-specific CD8+ T cell response in mice treated with cisplatin and E7 long peptide, while administration of anti-TNF-a suppressed the generation of such immune responses (Fig. 3d, e). When characterizing the TME following various treatments, increase in M1 macrophage and CD8+ T cell population and a decrease in M2 macrophage and MDSCs population were observed in mice treated with cisplatin, E7 long peptide, and anti-TGF-f3 as compared to those treated with cisplatin and E7 long peptide only, while the reverse trends were observed in those treated with cisplatin, E7 long peptide, and anti-TNF-a (Supplementary Fig. 6a–c). Anti-TGF-f3 treatment following cisplatin and E7 long peptide administration also decreased the level of TGF-f3 both systemically and within the TME while increasing the level of TNF-a, whereas the opposite effects were observed for anti-TNF-a administration (Supplementary Fig. 6d, e). These data demonstrate that the neutralization of TGF-f3 cytokine can induce similar therapeutic effects as AnxA5 treatment following chemotherapy and tumor antigen-specific vaccination and alleviate the immune suppression within the TME, while the neutralization of TNF-a results in the opposite effects, supporting the notion that the induction of TGF-f3 downregulation and TNF-a upregulation in the TME is a

potential mechanism by which AnxA5 exerts its immunostimulating function.

**Comparison of AnxA5 to other immune checkpoint inhibitors.** Combination treatment with antigen-specific immunotherapy and immune checkpoint inhibitors has been a promising emerging approach for enhancing the resultant immunogenicity and efficacy of immunotherapy (for review see ref. [31]). Due to the demonstrated potential of AnxA5 as an immune checkpoint inhibitor against PS-mediated immune suppression, we sought to assess the potency of AnxA5 administration as compared to the utilization of other reported immune checkpoint inhibitors by treating TC-1 tumor-bearing C57BL/6 mice that received cisplatin and/or E7 long peptide injection with further administration of AnxA5, anti-TGF-β, anti-PD-1, anti-PD-L1, or anti-TIM-3 (Fig. 4a). Administration of various immune checkpoint inhibitor following cisplatin treatment and E7 long peptide vaccination resulted in comparable levels of tumor control and prolonged mouse survival as compared to TC-1 tumor-bearing mice treated with cisplatin and E7 long peptide only (Fig. 4b, c). Tumor-bearing mice treated with cisplatin, E7 long peptide, and various immune checkpoint inhibitors also generated stronger systemic and tumor-infiltrating E7-specific CD8+ T cell responses compared to those treated with cisplatin and E7 long peptide only (Fig. 4d, e). Interestingly, treatment with AnxA5 resulted in the generation of slightly stronger systemic and tumor-infiltrating E7-specific CD8+ T cell responses compared to treatment with other checkpoint inhibitors (Fig. 4d, e). Together, these data demonstrate that AnxA5 treatment has comparable potency in enhancing the efficacy and immunogenicity of antitumor therapy compared to that of other immune checkpoint inhibitors.

**Antitumor effects of Annexin A5-peptide fusion protein.** Increasing evidence has highlighted the importance of infiltrating, local CD8+ T cell response in the treatment of cancer (for review see refs. [32–34]). Particularly, the presence of abundant and immunogenic tumor antigens within the TME is a crucial factor for the generation of effective, localized antitumor immune responses (for review see ref. [35]). We administered the E7 long peptide into TC-1 tumor-bearing mice via the intratumoral route, since we previously demonstrated that intratumoral administration of a therapeutic cancer vaccine results in the generation of more potent antigen-specific antitumor immune responses compared to systemic vaccination[36]. However, many intratumoral vaccinations are invasive in nature and less ideal for clinical application[37–42], which emphasizes the need to explore alternative methods that can generate strong, local antitumor immune responses via systemic administration. Due to the high affinity binding between AnxA5 and PS exposed on the surface of apoptotic and tumor cells, AnxA5 has been shown to possess tumor homing capabilities, and has been utilized as guiding and labeling tools for imaging tumor cell apoptosis[43]. We further confirmed the rapid and concentrated accumulation of Gaussia luciferase—AnxA5 fusion protein (AnxA5-Gluc) within the TME in vivo when administered following chemotherapy (Supplementary Fig. 7). We thus reasoned that AnxA5 can serve not only as an immune checkpoint inhibitor for tumor treatment, but also as a potent guiding molecule to home vaccine incorporated tumor antigens to the tumor.

To test our hypothesis, we fused the DNA sequence of murine H2-Db restricted epitope of HPV16-E7 (aa49-57: RAHYNIVTF) to that of AnxA5 to generate AnxA5-E7 fusion protein (Supplementary Fig. 8) and assessed the therapeutic efficacy of systemic AnxA5-E7 administration as compared to systemic administration of AnxA5 only or E7aa49-57 peptide only in TC-1

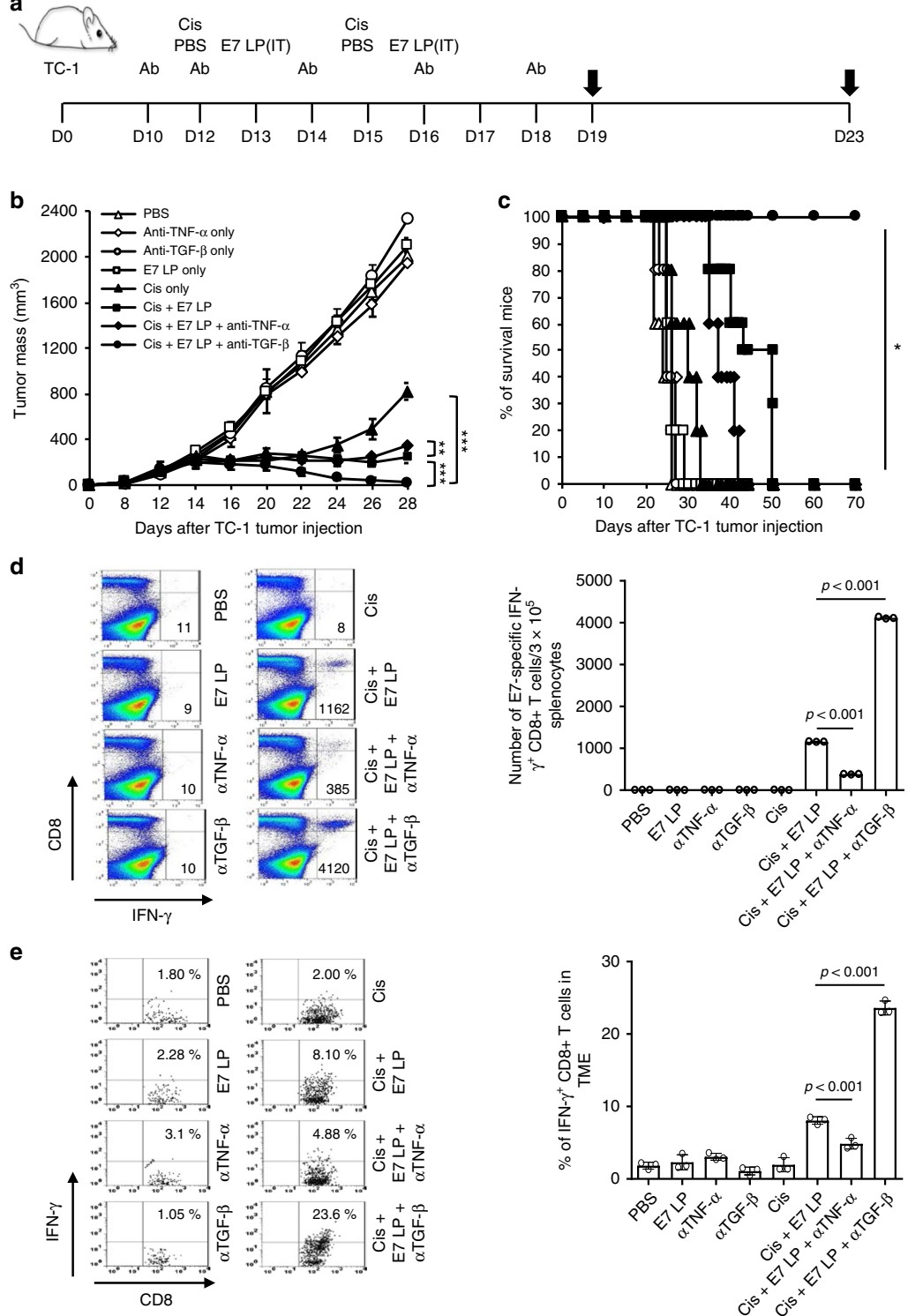

tumor-bearing mice treated with or without cisplatin (Fig. 5a). Tumor-bearing mice treated with cisplatin and AnxA5-E7 demonstrated significantly better tumor control and prolonged survival compare to those in other treatment groups (Fig. 5b, c). Furthermore, mice treated with cisplatin and AnxA5-E7 generated both strongest systemic and tumor infiltrating E7-specific CD8+ T cell responses (Fig. 5d, e). When compared to concomitant systemic AnxA5 protein and intratumoral E7 long peptide administration following cisplatin chemotherapy,

systemic AnxA5-E7 fusion protein administration following cisplatin chemotherapy resulted in more effective antigen presentation by DCs in the tumor draining lymph node (Supplementary Fig. 9), which translated into better tumor control, prolonged mouse survival, and the generation of stronger tumor-specific CD8+ T cell responses (Supplementary Fig. 10). The therapeutic antitumor effects generated by AnxA5-E7 vaccination following cisplatin treatment is CD8+ T cell dependent, as the administration of anti-CD8 neutralizing

**Fig. 3 Antitumor effect of anti-TGF-β or anti-TNF-α antibody treatment.** C57BL/6 mice were injected with $2 \times 10^5$ TC-1 cells/mice subcutaneously on day 0. Mice were then treated intraperitoneally with 200jig/mice TGF-3 or TNF-α neutralizing antibody on day 10, 12, 14, 16, and 18, intraperitoneally with 5 mg/kg Cisplatin on days 12 and 15, and/or intratumorally with 20jig/mice of E7 long peptide on days 13 and 16. The treatment groups are as follows: opened triangle - PBS only; opened sphere - anti-TNF-α only - opened circle, TGF-3 only - opened square, E7 long peptide only - closed triangle, cisplatin only - closed square, cisplatin and E7 long peptide - closed sphere, cisplatin, E7 long peptide, and anti-TNF-α; closed circle—cisplatin, E7 long peptide, and anti-TGF-f3. **a** Schematic diagram. **b** Line graph depicts TC-1 tumor growth in different treatment groups over time ($n = 10$). P-values were determined by one-way ANOVA and Turkey's test. (**c**) Kaplan–Meier survival analysis of TC-1 tumor-bearing mice in different treatment groups ($n = 10$), and the overall P-value was calculated by the log-rank test. **d**, **e** On days 19 and 23, tumor tissues and spleens of TC-1 tumor-bearing mice in different treatment groups were harvested and analyzed for CD8+IFN-γ+ T cells by flow cytometry analysis, respectively. **d** Representative flow cytometry analysis and bar graph depicting the abundance of CD8+IFN-γ+ T cells in splenocytes of TC-1 tumor-bearing mice in different treatment groups ($n = 3$). (**e**) Representative flow cytometry analysis and bar graph depicting the abundance of CD8+IFN-γ+tumor-infiltrating T cells in TC-1 tumor-bearing mice in different treatment groups ($n = 3$). The error bars indicate mean ± SD. For **d**, **e**, P-values were analyzed by Student's t test. The results are representative of one of three independent experiments. Source data are provided as a Source Data file.

antibody abolished the ability of cisplatin and AnxA5-E7 treatment mice to control TC-1 tumor growth (Supplementary Fig. 11).

**Assess general applicability of AnxA5-antigen peptide fusion.** We sought to evaluate the applicability of AnxA5-antigenic peptide fusion protein as a treatment strategy following a different chemotherapeutic agent, as well as against a different tumor model. We first evaluated the therapeutic antitumor efficacy of AnxA5–antigenic peptide fusion protein following the administration of doxorubicin chemotherapy. Similar to the experiments utilizing cisplatin chemotherapy, TC-1 tumor-bearing mice treated with doxorubicin and AnxA5-E7 generated strongest E7-specific CD8+ T cell response as well as potent tumor control and prolonged survival as compared to those in other treatment groups (Supplementary Fig. 12).

We then evaluated the therapeutic antitumor efficacy of AnxA5-antigenic peptide fusion protein using the CT26 murine colorectal cancer model. We fused AnxA5 with the AH5 peptide sequence (SPSYAYHQF), a reported tumor antigenic epitope of CT26, to generate the AnxA5-AH5 fusion protein. We then compared the therapeutic antitumor efficacy of AnxA5, AH5 peptide, and/or AnxA5-AH5 fusion protein following cisplatin chemotherapy in CT26 tumor bearing BALB/c mice (Fig. 6a). Similar to that observed in TC-1 tumor model, AnxA5-AH5 administration following cisplatin resulted in the best tumor control, prolonged survival of mice, as well as the generation of the strongest systemic and tumor infiltrating AH5-specific CD8+ T cell response as compared to those that received other treatment regimens (Fig. 6b–e).

When evaluating the TME of CT26 tumor-bearing mice following various treatments (Supplementary Fig. 13a), we saw an increase in the M2 macrophage and a decrease in the M1 macrophage population following cisplatin treatment, which was reversed by subsequent administration of AnxA5-AH5 (Supplementary Fig. 13b). Similar to that observed in TC-1 tumor model, AnxA5-AH5 administration following cisplatin significantly increased the CD8+ T cell and CD4+ T cell populations and reduced the Tregs and MDSCs population in the TME (Supplementary Fig. 13c). Likewise, we observed a significant increase in the level of TNF-α and a significant decrease in the level of TGF-β cytokines in the TME following cisplatin and AnxA5-AH5 treatment as compared to cisplatin treatment alone (Supplementary Fig. 13d).

Together, these data demonstrated that fusing antigenic peptides to AnxA5 is a viable strategy to enhance the therapeutic antitumor immune response following various chemotherapy and for the treatment of different tumors.

**Synergy between AnxA5 and other immune checkpoint inhibitors.** Recent studies have highlighted the therapeutic potential of rational combination of various immune checkpoint blockades for the generation of synergistic antitumor immunity[44,45]. Due to the potent therapeutic antitumor effects demonstrated by AnxA5 and antigenic peptide fusion protein following chemotherapy, we sought to assesses whether the therapeutic efficacy of AnxA5 and tumor antigenic peptide fusion protein can be further enhanced by other immune checkpoint inhibitors for the treatment of more advanced tumors. To do so, we treated TC-1 tumor-bearing mice with cisplatin and/or AnxA5-E7 with or without additional treatment with anti-PD-1, anti-PD-L1, or anti-TIM-3, using a delayed treatment schedule starting at 15 days after tumor challenge, when the tumors have reached the size of about 300 mm$^3$ (Fig. 7a). While the initiation of cisplatin and AnxA5-E7 treatment at 15 days after TC-1 tumor challenge could result in the initial control of TC-1 tumor, resurgence of tumor growth was observed at 35 days after tumor challenge, and only 40% of mice were still alive at 70 days after tumor challenge (Fig. 7b, c). In comparison, no recurrence of TC-1 tumor growth was observed in mice treated with cisplatin, AnxA5-E7, and anti-PD-1, anti-PD-L1, or anti-TIM-3 (Fig. 7b). In addition, all mice treated with cisplatin, AnxA5-E7, and additional immune checkpoint inhibitors survived for at least 70 days after TC-1 tumor challenge (Fig. 7c). Compared to those in TC-1 tumor-bearing, cisplatin and AnxA5-E7 treated mice, combination treatment of cisplatin, AnxA5-E7, and additional immune checkpoint inhibitor generated significantly stronger E7-specific CD8+ T cell response in TC-1 tumor-bearing mice (Fig. 7d). These data support the rational combination of AnxA5-E7 with additional immune checkpoint blockade therapy for the generation of enhanced therapeutic antitumor immunity following chemotherapy.

**Discussion**

This study demonstrated the ability to inhibit chemotherapy-induced immune suppression in the TME through AnxA5 administration. Particularly, we showed that systemic administration of AnxA5 following chemotherapy can enhance the immunogenicity of local tumor antigens. We also demonstrated that AnxA5 can be fused with antigenic peptides and serve as a guide molecule to concentrate antigenic peptides within the TME. In addition, AnxA5 treatment can further synergize with other immune checkpoint inhibitors for enhanced therapeutic potency. We confirmed that the AnxA5 proteins used in the current study, which are produced via *E.coli* based protein purification method, are not contaminated with endotoxin (Supplementary Fig. 14).

Our findings further demonstrate the role of PS as an important immune checkpoint that restricts the generation of effective antitumor immunity. Furthermore, unlike many other immune

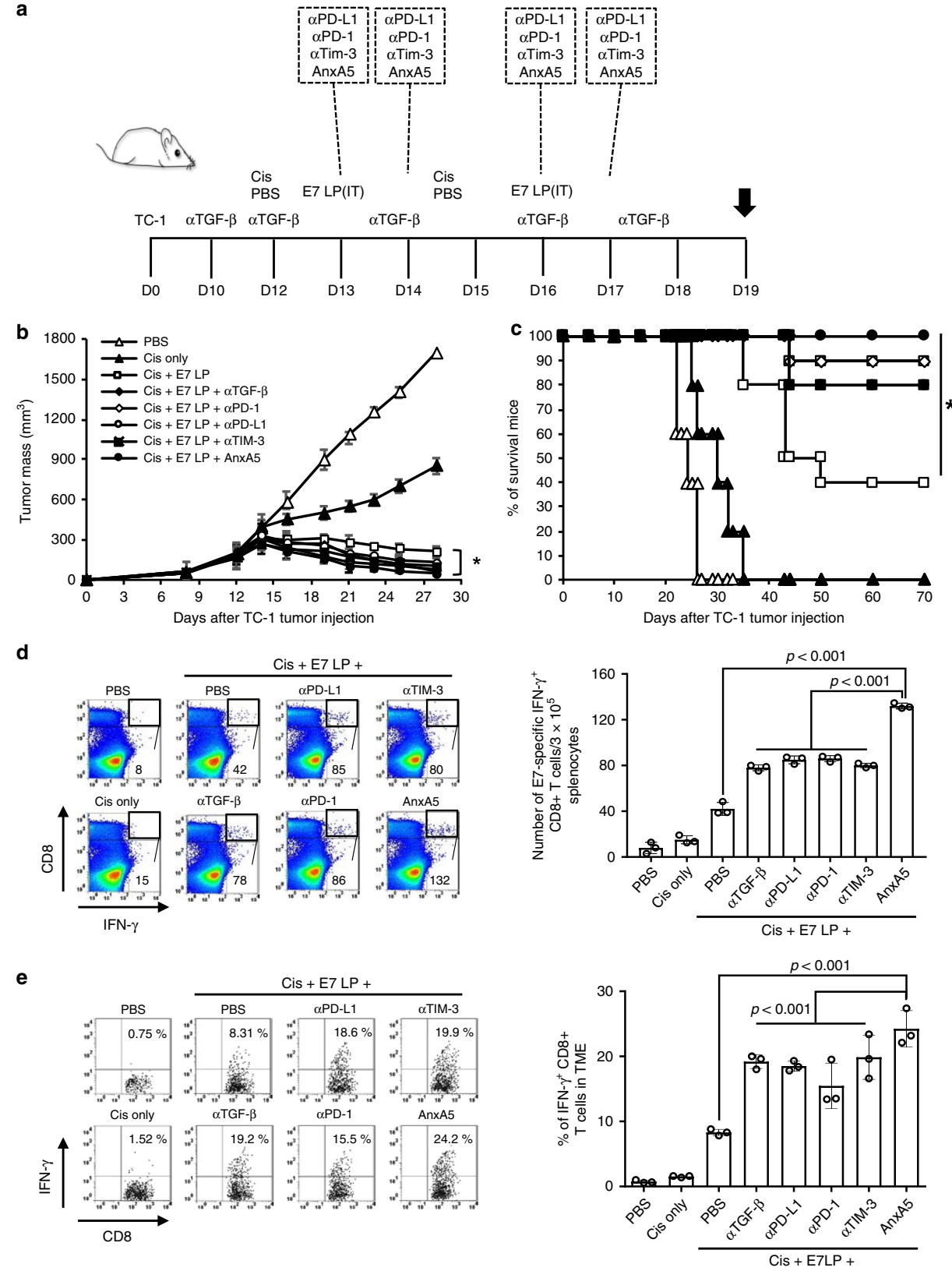

checkpoints, PS has been suggested as a global immune checkpoint, inducing immune suppressive signals via multiple pathways (reviewed in ref. [5]). Here we showed that interaction of PS promotes TGF-β secretion, inhibits TNF-α production by innate immune phagocytes inside the TME (Fig. 2 & Supplementary Fig. 3), and suppresses proper function of CD8+ T cells (Supplementary Fig. 5). Furthermore, we found that cisplatin-induced tumor apoptosis contributes to the upregulation of PD-L1 in

**Fig. 4 Antitumor efficacy of Annexin A5 versus other immune checkpoint inhibitors.** C57BL/6 mice were injected with $2 \times 10^5$ TC-1 cells/mouse subcutaneously on day 0. Mice were then treated intraperitoneally with 200jig/mice α-TGF-f3 on day 10, 12, 14, 16, and 18, intraperitoneally with 5 mg/kg Cisplatin on days 12 and 15, intravenously with 200jig/mice of Annexin A5 proteins on days 13, 14, 16, and 17, intraperitoneally with 200jig/mice of α-PD-1, α-PD-L1, or α-TIM-3 on days 13, 14, 16, and 17, and/or intratumorally with 20jig/mice of E7 long peptide on days 13 and 16. PBS and irrelevant IgG were used as controls. The treatment groups are as follows: opened triangle - PBS only; closed triangle - cisplatin only; opened square—cisplatin and E7 long peptide; closed sphere—cisplatin, E7 long peptide, and anti-TGF-f3; opened sphere—cisplatin, E7 long peptide, and anti-PD-1; opened circle—isplatin, E7 long peptide, and anti-PD-L1; closed square—cisplatin, E7 long peptide, and anti-TIM-3; closed circle—cisplatin, E7 long peptide, and Annexin A5. **a** Schematic diagram. **b** Line graph depicts TC-1 tumor growth in different treatment groups over time ($n = 10$). P-values were determined by one-way ANOVA and Turkey's test. **c** Kaplan–Meier survival analysis of TC-1 tumor-bearing mice in different treatment groups ($n = 10$), and the overall P-value was calculated by the log-rank test. **d**, **e** On day 19, spleens and tumor tissues of TC-1 tumor-bearing mice in different treatment groups were harvested and analyzed for CD8+IFN-γ+ T cells by flow cytometry analysis. **d** Representative flow cytometry analysis and bar graph depicting the abundance of CD8+IFN-γ+ T cells in splenocytes of TC-1 tumor-bearing mice in different treatment groups ($n = 3$). **e** Representative flow cytometry analysis and bar graph depicting the abundance of CD8+IFN-γ+tumor-infiltrating T cells in TC-1 tumor-bearing mice in different treatment groups ($n = 3$). The error bars indicate mean ± SD. For **d**, **e**, P-values were analyzed by Student's t test. The results are representative of one of three independent experiments. Source data are provided as a Source Data file.

TME (Fig. 2d). It has been previously reported that many tumor cells and immunosuppressive cells within the TME upregulate the expression of receptor tyrosine kinases Tyro3, Axl, and Mertk (collectively known as TAM receptors), which can interact with PS to stimulate the expression of PD-L1[46]. Thus, our results further support the notion that PS may act akin or upstream of PD-L1, CTLA-4, TGF-β, and TIM-3, representing an important immune checkpoint to be targeted for enhancing the efficacy of antitumor immunotherapy. Furthermore, a previous study has reported a population of TIM-3 expressing Tregs that can interact with PS to exert potent suppressive effects against effector T cells[47], and it has been recently demonstrated that the hypoxic state of the TME induces Tregs to undergo apoptosis, which exerts a strong immunosuppressive effect within the TME, partly via the subsequent uptake and processing by phagocytic macrophages[48]. While this phenomenon is not explored in the current study, it is possible that AnxA5 treatment exerts its immunostimulating effects, at least partially, by suppressing the effects of TIM-3 expressing Tregs or by inhibiting the interactions between apoptotic Tregs and phagocytes in the TME.

The association between PS expression and immunosuppressive apoptotic clearance has been shown to have critical roles not only in the shaping of an immunosuppressive TME, but also in the context of autoimmune disorder development and during pathogen infections. The natural, intended role of PS externalization by apoptotic cells is to facilitate the effective clearance of apoptotic cell debris while inducing the production of immunosuppressive cytokines to prevent the induction of unwanted inflammatory responses during the homeostatic erythrocytosis of aged, decaying cells. As such, defects in the PS signaling pathway has been associated with undesired activation of inflammatory and immune responses. Indeed, previous studies have shown that loss of PS–PS receptor signaling in mice led to the emergence of autoimmune diseases resembling systemic lupus erythematous, and that defects in PS—PS receptor signaling have been identified in some portion of patients with systemic lupus erythematosus[49]. On the other hand, due to the ability of PS in promoting the immunosuppressive uptake by phagocytes, PS has also been utilized by various pathogens including virus, bacteria, and parasites to promote their infectivity while inhibiting the host's immune system to mount an effective response for pathogen clearance (for review see ref. [5]). It has been reported that various viral pathogens engage in a process known as viral apoptotic mimicry, in which the virus will display PS on its surface to resemble apoptotic debris to promote its uptake by phagocytes for subsequent infection while inducing the production of immunosuppressive cytokines by the phagocytes to prevent the generation of viral-targeting immune response[50]. Similarly, various bacteria and

parasites were shown to either directly express PS on its cell surfaces or promote the upregulation of surface PS by the infected and/or bystander cells to inhibit the effective immunologic clearance of infection (for review see ref. [5]). As such, the appropriate manipulation of PS signaling, such as the use of AnxA5, may also serve as a rational approach for the control of these diseases.

Although our paper focuses on the ability of AnxA5 to act as an immune checkpoint inhibitor against PS-induced tumor immune suppression, our findings also support the importance of potent, localized antigen-specific immune responses for effective cancer treatment. While AnxA5 administration alone following cisplatin treatment can rescue the immunosuppressive effects within the TME (Fig. 2 and Supplementary Fig. 13), cisplatin and AnxA5 treatment is not effective at controlling the growth of tumor and prolonging mouse survival (Figs. 1, 5, and 6). The correlation of strong, tumor infiltrating, local CD8+ T cell responses and the regression of cancer has been demonstrated in previous studies[51,52], and strategies to enhance the generation of tumor infiltrating lymphocytes have become an emerging focus of immunotherapeutic development[53–56]. Here we showed that due to the concentrated PS expression within the TME, AnxA5 can serve not only as an immune checkpoint inhibitor to combat tumor immune suppression, but also as a homing molecule to target fused antigenic peptides to TME and enhance the localized antigen-specific antitumor immunity (Figs. 5–7). We hypothesize that similar approaches are also applicable against other immune checkpoint, and that tumor antigen fused immune checkpoint inhibitors may be a promising approach to not only abolish the immune suppressive effects within the TME, but also stimulate the generation of enhanced tumor-specific T cell responses.

While our results support the therapeutic potential of PS-targeting immune checkpoint blockade by AnxA5, we recognize certain limitations. Particularly, the current study did not compare the efficacy of AnxA5 versus several reported anti-PS antibodies[14–16]. While AnxA5 is expected to possess a stronger binding affinity than anti-PS antibodies[57], AnxA5-based treatment may suffer from the short physical half-life of AnxA5[58]. Short physical half-life is a critical issue barring the feasibility of many therapeutic agents. Many strategies have been examined to extend the half-life of therapeutic agents, including fusing therapeutic peptide or protein molecules to the fragment crystallizable (Fc) region of antibody or to albumin binding domains, thereby undergoing transcytotic recycling via an Fc receptor (FcRn)-mediated manner (reviewed in refs. [59,60]). Further fusing AnxA5-antigenic peptide fusion protein to albumin or Fc region may be an effective approach to further enhance the half-life and therefore the therapeutic activity of AnxA5 -based immunotherapeutic

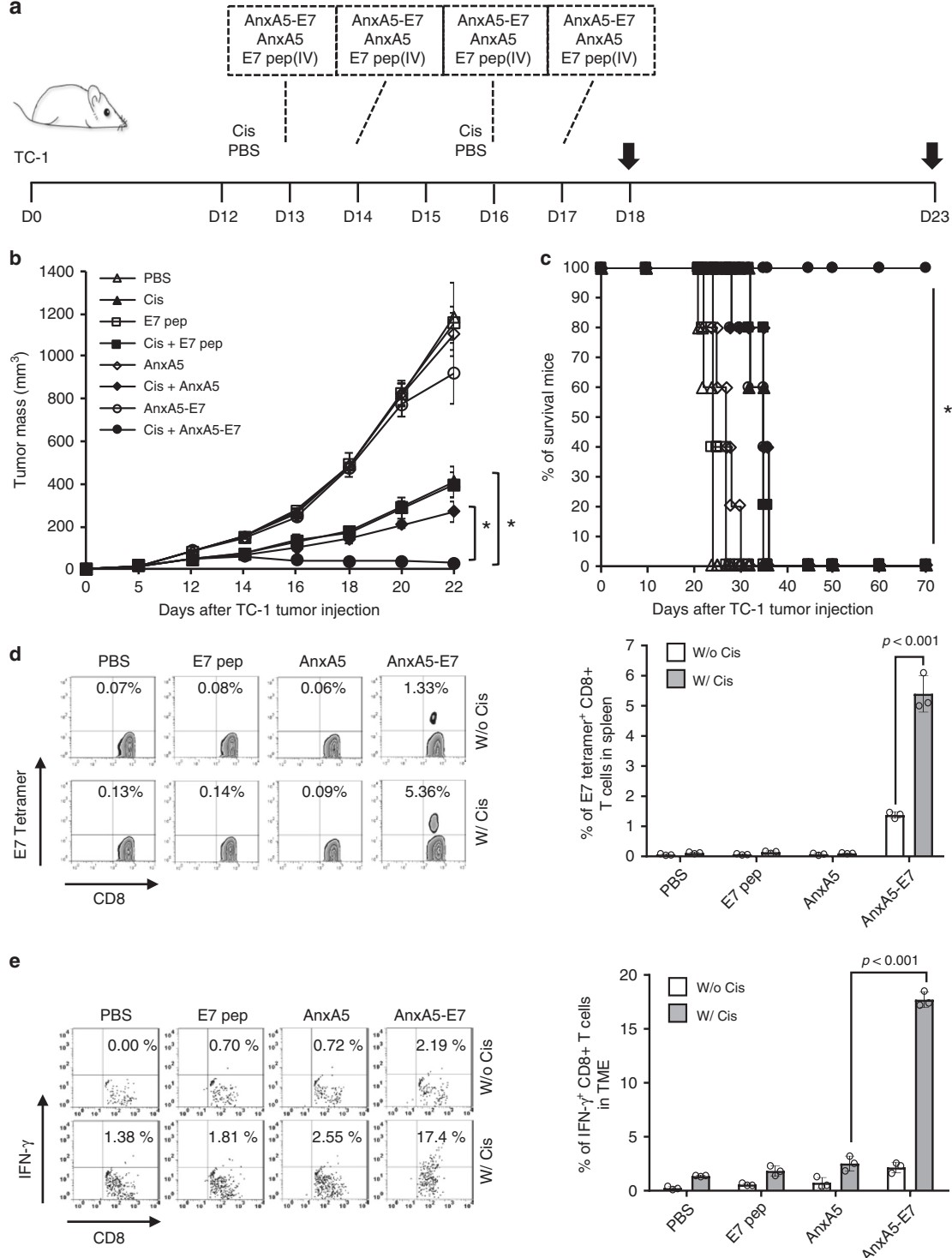

and are warranted in future studies. Finally, while we demonstrated that AnxA5-antigenic peptide fusion proteins have similar therapeutic antitumor effects against TC-1 and CT26 tumor models, further evaluations in additional tumor models is warranted to assess any potential treatment variations.

This study reports the potential of targeting tumor PS as an effective approach to enhance the immunogenicity of the TME and immunotherapy for the generation of potent antitumor immunity. Due to the current application of AnxA5 as tumor imaging tools in clinical studies[61], such approaches have high clinical feasibility and translational potential and may lead to the development of novel and readily adaptable immunotherapeutic for the treatment of cancer.

## Materials and methods

**Mice**. Six- to eight-week-old female C57BL/6 and BALB/c mice were purchased from the Orient Bio Inc. All animal procedures were performed according to protocol approved by the Institutional. Animal Care and Use Committee (IACUC) of Konkuk University (KU16043).

**Cells**. The HPV16 E6 and E7 oncoproteins expressing murine tumor model, TC-1 tumor cells, were generated by cotransformation of primary C57BL/6 mice lung epithelial cells with HPV-16 E6, E7, and an activated *ras* oncogene[25]. CT26 murine

**Fig. 5 Antitumor effects of recombinant Annexin A5-E7 fusion protein.** C57BL/6 mice were injected with $2 \times 10^5$ TC-1 cells/mouse subcutaneously on day 0. Mice were then treated intraperitoneally with 5 mg/kg Cisplatin on days 12 and 15, intravenously with 200jig/mice of Annexin A5-E7 fusion protein, 200jig/mice of Annexin A5 proteins, and/or 3.5jig/mice of E7 long peptide on days 13, 14, 16, and 17. The treatment groups are as follows: opened triangle —PBS only; closed triangle—cisplatin only; opened square—E7 peptide only; closed square—cisplatin and E7 peptide; opened sphere—Annexin A5 only; closed sphere—cisplatin and Annexin A5; opened circle—Annexin A5-E7peptide only; closed circle—cisplatin and Annexin A5-E7 peptide. **a** Schematic diagram. **b** Line graph depicting TC-1 tumor growth in different treatment groups over time ($n = 10$). $P$-values were determined by one-way ANOVA and Turkey's test. **c** Kaplan–Meier survival analysis of TC-1 tumor-bearing mice in different treatment groups ($n = 10$), and the overall $P$-value was calculated by the log-rank test. **d, e** On days 18 and 23, tumor tissues and spleens of TC-1 tumor-bearing mice in different treatment groups were harvested and analyzed for CD8+IFN-γ+ or CD8+E7tetramer+ T cells by flow cytometry analysis, respectively. **d** Representative flow cytometry analysis and bar graph depicting the abundance of CD8+E7tetramer+ T cells in splenocytes of TC-1 tumor-bearing mice in different treatment groups ($n = 3$). **e** Representative flow cytometry analysis and bar graph depicting the abundance of CD8+IFN-γ+tumor-infiltrating T cells in TC-1 tumor-bearing mice in different treatment groups ($n = 3$). The error bars indicate mean ± SD. For (**d**, **e**), $P$-values were analyzed by Student's $t$ test. The results are representative of one of three independent experiments. Source data are provided as a Source Data file.

colon carcinoma cells were purchased from ATCC. E7(aa49-57)-specific CD8+ T cells were generated by immunizing C57BL/6 (H-2[b]) mice with $10^7$ p.f.u. of Sig/ E7/LAMP-1 vaccinia virus, harvesting the splenocytes at day 8 after immunization, and stimulating with autologous, irradiated splenocytes pulsed with 1jiM E7 peptide (aa49-57) and 30units/ml of recombinant human IL-2[9]. OVA-specific CD8+ T cells (OT-1) were generated by stimulating splenocytes harvested from OT-1 transgenic mice with irradiated splenocytes pulsed with 1jiM of OVA peptide (aa257-264) at the presence of 20units/ml of IL-2[9]. These cell lines were cultured in vitro in RPMI-10 medium (Biowest; RPMI 1640 supplemented with 10% fetal bovine serum, 50units/ml of penicillin/streptomycin, 2mM L-glutamine, 1 mM sodium pyruvate, and 2 mM non-essential amino acids) at 37 °C with 5% CO2.

To obtain CD8+ T cells from treated TC-1 tumor-bearing mice, three C57BL/6 mice were injected with $2 \times 10^5$ TC-1 cells/mouse subcutaneously on day 0. Mice were then treated intraperitoneally with 5 mg/kg Cisplatin on days 12 and 15, intravenously with 200jig/mice of AnxA5-E7 fusion protein on days 13, 14, 16, and 17. On days 23, spleens of TC-1 tumor-bearing mice were harvested and CD8+ T cells were purified using CD8a+ T Cell Isolation Kit (MACS Miltenyi Biotec, Germany).

BMDCs or BMDM were obtained by culturing monocytes harvested from the bone marrow of C57BL/6 mice in RPMI-10 medium supplemented with 2 mM Granulocyte-macrophage colony-stimulating factor (GM-CSF) or DMEM/F12 medium (Biowest) supplemented with 10% fetal bovine serum, 50 units/ml of penicillin/streptomycin, 15 mM Hepes and 50 ng/ml macrophage colony-stimulating factor (M-CSF), respectively, at 37 °C with 5% CO₂. The monocytes were incubated for 6 days before experimentation.

**Protein production and purification.** To generate pET28-AnxA5, human annexin V was first amplified by PCR using pet12a-PAP1 (from Addgene) as a template and the following set of primers:
5′- TTTGGATCCATGGCACAGGTTCTCAGAGG -3′ and
5′- AAAGAATTCGTCATCTTCTCCACAGAGCA -3′. The PCR product was cloned into the BamHI and EcoRI sites of the pET28a vector.

To generate pET28- AnxA5-E7, AnxA-E7was first amplified by PCR using pet12a-PAP1 as a template and the following set of primers: 5′- TTTGGATCCAT GGCACAGGTTCTCAGAGG - 3′ and
5′-AAAGAATTCAAAGGTTACAATATTGTAATGGGCTCTGTCATCTTC TCCACAGAGC A -3′. The PCR product was cloned into the BamHI and EcoRI sites of the pET28a vector.

To generate pET28-AnxA5-AH5, oligos (AATTCTCCCCCTCCTACGCCTACCACCAGTTCTAAC and TCGAGTTAGAACTGGTGGTAGGCGTAGGAGGGGGAG) were cloned into EcoRI and XhoI sites of pET28- AnxA5.

To generate pET28- Gluc, Gaussia luciferase was first amplified by PCR using phGluc (from Addgene) as a template and the following set of primers:
5′- AAAGAATTCGAGGCCAAGCCCACCGAGAAC -3′ and
5′- TTTCTCGAGGTCACCACCGGCCCCCTTGA -3′. The PCR product was cloned into the EcoRI and XhoI sites of the pET28a vector.

To generate pET28-AnxA-Gluc, the Gaussia luciferase PCR product was cloned into the EcoRI and XhoI sites of the pET28- AnxA5.

The DNA plasmids were confirmed by sequencing and transformed into Escherichia coli (BL21(DE3)). The selected colony was cultured in 5 mL Luria–Bertani (LB) liquid medium containing Kanamycin (50 μg/mL) overnight at 37 °C on a shaking incubator, then transferred to 250 mL of fresh medium (with the antibiotic) and incubated for another 2 h until the optical density (OD 600) of the cultured cells reached approximately 0.6. Expression of the fusion protein was induced with 1 mM isopropyl-b-D-thiogalactopyranoside (IPTG) at 37 °C for 5 h. The cultured cells were harvested by centrifugation at 2460xg for 10 min at 4 °C. The pellet was washed with phosphate buffered saline (PBS) two times and then suspended in bacteria lysis buffer (SoluLyse Reagent for Bacteria, Genlantis) containing lysozyme (100jig/ml) (Gibco BRL) and deoxynuclease (Dnase)I (100U/ml) (Invitrogen). The suspension was incubated for 2 h at room temperature

with stirring, followed by centrifugation at 10241xg for 15 min. The clear supernatant (soluble fraction) was collected and the recombinant protein was purified by Ni+ affinity chromatography (Ni-NTA agarose, Qiagen) according to the manufacturer's protocol. The eluted protein was collected and analyzed using 10-15% gradient SDS–PAGE and coomassie brilliant blue staining. The purity of proteins was characterized by limulus amoebocyte lysate (LAL) (Lonza) and Picogreen assays (Invitrogen). The endotoxin level of each protein was less than 0.01EU/mg, and the bacterial DNA level was 0.1 ng/mg of protein in independent preparations.

**In vivo tumor treatment experiments.** For in vivo tumor treatment experiment in TC-1 tumor model, $2 \times 10^5$ TC-1 cells were injected subcutaneously into C57BL/6 mice (10 per group). 5 mg/kg Cisplatin (Sigma-Aldrich, Germany) or 10 mg/kg Doxorubicin (Sigma-Aldrich, Germany) was administered via intraperitoneal injection on days 12 and 15. 20jig/mice of E7 42-63 long peptide (AGQAEP-DRAHYNIVTFCCKCDS) were injected intratumorally on days 13 and 16. 3.5jig/ mice of E7 49-57 peptide (RAHYNIVTF) were injected intravenously into the lateral tail vein on days 13, 14, 16, and 17. 200jig/mice of AnxA5 or AnxA5-E7 were injected intravenously into the lateral tail vein on days 13, 14, 16, and 17. 200jig/mice of anti-TGF-β (clone 1D11.16.8) or anti-TNF-α (clone XT3.118) antibodies (BioXcell, USA) were injected intraperitoneally on days 10, 12, 14, 16, and 18. 200jig/mice of anti-PD-1 (clone RMP1-14), anti-PD-L1 (clone 10 F.9G2), or anti-TIM-3 (clone RMT3-23) antibodies (BioXcell, USA) were injected intra-peritoneally on days 13, 14, 16, and 17.

For in vivo tumor treatment experiments in CT26 tumor model, $5 \times 10^5$ CT 26 cells were injected subcutaneously into BALB/c mice (10 per group). 5 mg/kg Cisplatin was administered via intraperitoneal injection on days 12 and 15. 3.5jig/ mice of AH5 peptide (SPSYAYHQF) were injected intravenously into the lateral tail vein on days 13, 14, 16, and 17. 200jig/mice of AnxA5 or AnxA5-AH5 were injected intravenously into the lateral tail vein on days 13, 14, 16, and 17.

For in vivo CD8 depletion experiment, C57BL/6 mice (10 per group) were injected subcutaneously with $2 \times 10^5$ TC-1 cells on day 0. 5 mg/kg Cisplatin was administered via intraperitoneal injection on days 12 and 15. 200jig/mice of AnxA5-E7 were injected intravenously into the lateral tail vein on days 13, 14, 16, and 17. 100jig/mice of anti-CD8 (clone 2.43) or control IgG (Rat IgG2b, clone LTF-2) antibodies (BioXcell, USA) were injected intraperitoneally daily from day 12 to day 20.

For advanced tumor treatment experiment, $2 \times 10^5$ TC-1 cells were injected subcutaneously into C57BL/6 mice (10 per group). 5 mg/kg cisplatin was injected intraperitoneally on days 15 and 18. 200jig of AnxA5-E7 protein was injected intravenously on days 16, 17, 19, and 20. 200jig/mice anti-PD-1, anti-PD-L1, or anti-TIM3 antibodies were injected intraperitoneally on days 21, 23, and 25.

For all experiments, PBS injections were used as negative control.

Following tumor challenge, mice were monitored for evidence of tumor growth by palpation and inspection twice a week until death.

**In vitro T cell activation and proliferation assay.** Splenocytes of C57BL/6 mice were harvested and labeled via incubation with 5uM CFSE solution (Thermo Fisher, USA) for 15miuntes at 37 °C. $2 \times 10^6$ CFSE labeled splenocytes were then incubated with anti-CD3α mAb (Invitrogen, USA) or PMA (Sigma-Aldrich, Germany) plus Ionomycin (Sigma-Aldrich, Germany) in the presence of Phosphati-dylserine and/or AnxA5 protein. 24 h after incubation, the supernatants were harvested and assessed for IFN-γ cytokine level using ELISA. 3 days after incu-bation, the number of CFSE+ cells were analyzed using flow cytometry (Supple-mentary Fig. 18c).

For OVA-specific T cell activation experiment, splenocytes from OT-1 mouse were harvested and stimulated 1jig/ml OVA peptide (SIINFEKL) for 7 days. CD8+ T cells were isolated from splenocytes using CD8+ T cell MicroBeads (MACS Miltenyi Biotec, Germany), incubated with 5uM CFSE solution (Thermo Fisher, USA) for 15miuntes at 37 °C. $1 \times 10^6$ CFSE labeled CD8+ T cells were then

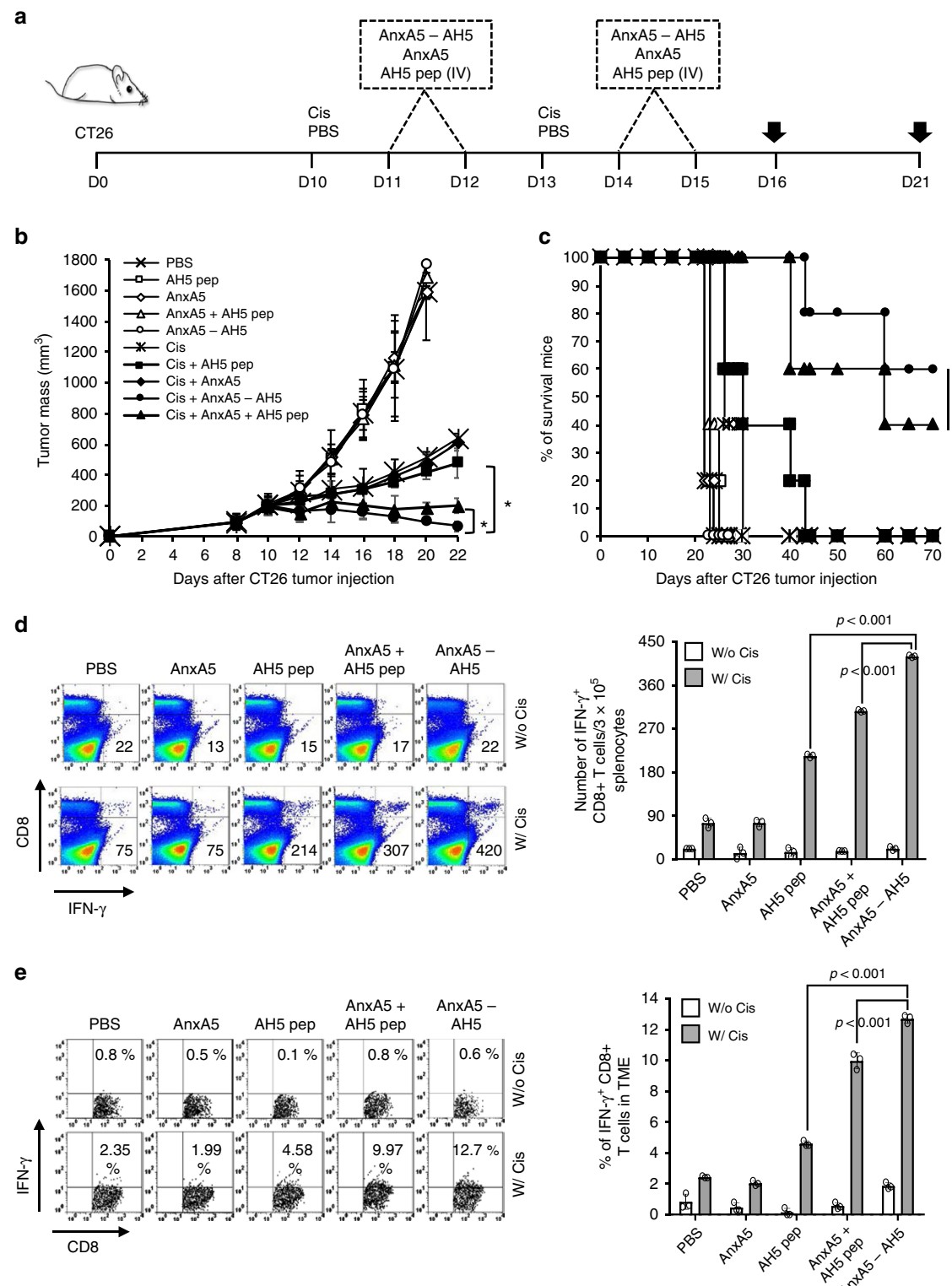

co-cultured with OVA expressing TC-1 cells treated with or without cisplatin, with or without concurrently 20jig/ml AnxA5 protein treatment. 24 h after incubation, the supernatants were harvested and assessed for IFN-γ cytokine level using ELISA. 3 days after incubation, the number of CFSE+ cells were analyzed using flow cytometry (Supplementary Fig. 18c).

**Analysis of systemic antigen-specific CD8+ T cell response.** For tetramer staining, PBMCs (peripheral blood mononuclear cells) were harvested 1 week after the last protein injection from the ventral arteries of mice tails into 1.5 ml EDTA

-coated Eppendorf tubes, then treated with ACK solution (Quality Biological, Gaithersburg, MD, USA) for RBC lysis[54]. After RBC lysis, single cells were stained with Phycoerythrin-labeled H-2Db HPV16 E7 (RAHYNIVTF) tetramer (Beckman Coulter, Hialeah, FL) and anti-CD8 antibody (Supplementary Fig. 17b).

For intracellular cytokine staining, splenocytes from each vaccination group were harvested 1 day or 1 week after the last protein injection. $5 \times 10^6$ pooled splenocytes from each vaccination group were incubated with 1jig/ml E7 49-57 peptide (RAHYNIVTF) or AH1 423–431 peptide (SPSYVYHQF) and 1ul/ml GolgiPlug (BD Cytofix/Cytoperm Kit) for 16 h. Cells were then harvested and

**Fig. 6 Antitumor effects of Annexin A5-AH5 fusion protein.** BALB/c mice were injected with $5 \times 10^5$ CT-26 cells/mouse subcutaneously on day 0. Mice were then treated intraperitoneally with 5 mg/kg cisplatin on days 10 and 13 and intravenously with 200jig/mice of AnxA5, 200jig/mice of AnxA5-AH5, and/or 3.5jig/mice of AH5 peptide on days 11, 12, 14, and 15. The treatment groups are as follows: cross—PBS only; opened square—AH5 peptide only; opened sphere—Annexin A5 only; opened triangle—Annexin A5 and AH5 peptide; opened circle—Annexin A5-AH5 only; star mark—cisplatin only; closed square—cisplatin and AH5 peptide; closed sphere—cisplatin and Annexin A5; closed circle—cisplatin and Annexin 5A-AH5; closed triangle—cisplatin, Annexin A5, and AH5 peptide. **a** Schematic diagram. **b** Line graph depicts CT-26 tumor growth in different treatment groups over time ($n = 10$). $P$-values were determined by one-way ANOVA and Turkey's test. **c** Kaplan–Meier survival analysis of CT-26 tumor-bearing mice in different treatment groups ($n = 10$), and the overall $P$-value was calculated by the log-rank test. **d–e** On days 16 and 21, tumor tissues and spleens of CT26 tumor-bearing mice in different treatment groups were harvested and analyzed for CD8+IFN-γ+ T cells by flow cytometry analysis, respectively. **d** Representative flow cytometry analysis and bar graph depicting the abundance of CD8+IFN-γ+ T cells in splenocytes of CT-26 tumor bearing mice in different treatment groups ($n = 3$). **e** Representative flow cytometry analysis and bar graph depicting the abundance of CD8+IFN-γ+tumor-infiltrating T cells in CT-26 tumor bearing mice in different treatment groups ($n = 3$). The error bars indicate mean ± SD. For (**d**, **e**), $P$-values were analyzed by Student's $t$ test. The results are representative of one of three independent experiments. Source data are provided as a Source Data file.

---

stained with FITC-labeled anti-CD8 and PE labeled anti-IFN- γ monoclonal antibodies (1:100 dilution).

All samples were analyzed by FACSCalibur flow cytometer. The samples were first gated for singlets by FSC-H and FSC-A as well as for living cells by side scatter area and forward scatter area, followed by analysis of gated lymphocyte populations, using the CellQuest software (Becton Dickinson, San Jose, CA) (Supplementary Fig. 15a).

**Tumor microenvironment cell analysis**. Tumors were surgically, dissected, washed twice with PBS, and digested using gentleMACS dissociator (Miltenyi Biotec, Germany) and MACS tumor dissociation kits (Miltenyi Biotec, Germany) with standard protocol. The digested tumor was filtered through a 100 µm cell strainer, followed by centrifugation. The cells were then washed twice using PBS. Remaining red blood cells were lysed using ACK solution.

To assess the presence of T cell population, cells were stained with PE-conjugated anti-CD8 (1:100 dilution) (53-6.7/biolegend)(Supplementary Fig. 16a) or CD4 antibodies (1:100 dilution) (GK1.5/invitrogen) (Supplementary Fig. 15d). To assess the MDSCs population, cells stained with PE-conjugated anti-CD11b (M1/70 /invitrogen) (1:100 dilution) and FITC-conjugated anti-Gr1 antibodies (RB6-8C5/invitrogen) (1:100 dilution) (Supplementary Fig. 16c). To assess M1 and M2 macrophage population, cells were stained with APC-conjugated F4/80(BM8/ invitrogen) (1:100 dilution)and FITC-conjugated CD206 antibodies (1:100 dilution) (C068C2/biolegend)(Supplementary Fig. 15c). To assess the presence of regulatory T cells, cells were stained with PE-cy7-conjugated anti-CD4(RM4-5/ invitrogen) (1:100 dilution) and APC-conjugated anti-CD25 antibodies (PC61.5/ invitrogen) (1:100 dilution) at 4 ℃ for 30 min, washed with PBS, incubated in Fixation/Permeabilization working solution at 4 ℃ for 20 min, and stained for PE-conjugated anti-Foxp3 antibodies (FJK-16S/invitrogen) (1:100 dilution) (Supplementary Fig. 16b). To assess the expression of PD-L1 by immune cells and tumor cells, cells were stained with FITC-conjugated anti-CD45 antibodies (30-F11/invitrogen) (1:100 dilution) and PE-conjugated anti-PD-L1 antibodies (10 F.9G2/invitrogen) (1:100 dilution) (Supplementary Fig. 17a). To assess the tumor-infiltrating antigen-specific CD8+ T cell population, cells were incubated with 1 µg/ml E7 49-57 peptide (RAHYNIVTF) and 1ul/ml GolgiPlug (BD Cytofix/ Cytoperm Kit) for 16 h followed by surface CD8 and intracellular IFN-γ antibodies (1:100 dilution).

All samples were analyzed by FACSCalibur flow cytometer. The samples were first gated for singlets by FSC-H and FSC-A as well as for living cells by side scatter area and forward scatter area, followed by analysis of gated lymphocyte populations, using the CellQuest software (Becton Dickinson, San Jose, CA) (Supplementary Fig. 15c–h).

**Splenic MDSC and Treg cell analysis**. Splenocytes of TC-1 tumor-bearing, cisplatin and/or AnxA5 treated C57BL/6 mice (5 mice per group) were harvested on day 18 following tumor challenge. To assess the MDSCs population, cells stained with PE-conjugated anti-CD11b (M1/70 /invitrogen) (1:100 dilution) and FITC-conjugated anti-Gr1 antibodies(RB6-8C5/invitrogen) (1:100 dilution). To assess the presence of regulatory T cells, cells were stained with PE-cy7-conjugated anti-CD4(RM4-5/invitrogen) (1:100 dilution) and APC-conjugated anti-CD25 antibodies (PC61.5/invitrogen) (1:100 dilution) at 4 ℃ for 30 min, washed with PBS, incubated in Fixation/Permeabilization working solution at 4 ℃ for 20 min, and stained for PE-conjugated anti-Foxp3 antibodies (FJK-16S/invitrogen) (1:100 dilution). All samples were analyzed by FACSCalibur flow cytometer. The samples were first gated for singlets by FSC-H and FSC-A as well as for living cells by side scatter area and forward scatter area, followed by analysis of gated lymphocyte populations, using the CellQuest software (Becton Dickinson, San Jose, CA).

**ELISA**. For in vivo cytokine analysis, harvested tumor tissues were chopped and resuspended on ice in RIPA protein extraction solution (50nmol/L Tris-Cl [Ph8.0], 150nmol/L NaCl, 1 mmol/L phenylmethylsulphonyl fluoride [PMSF], 0.1% sodium dodecyl sulfate [SDS], 1% Nonidet P-40 [NP-40] and 0.5 mmol/L EDTA) for 2 h and centrifuged at 12019xg for 15 min. Supernatant protein concentrations were determined by Bradford protein assay. The levels of TNF-ct and IL-10 cytokines and CCL2(MCP-1) chemokine in 1 mg of proteins from each treatment groups were quantified using a mouse Ready-Set-Go ELISA kit (Invitrogen, USA) following manufacturer's recommendations. The TGF-3 cytokine levels were measured used a mouse TGF-3 ELISA kit (R&D system, USA). All results were measured with a plate reader.

For in vitro cytokine analysis, $1 \times 10^5$ of TC-1 tumor cells were treated with or without 20 µg/ml of cisplatin for 6 h, and all tumor cells were harvested and washed twice with PBS. Apoptotic-tumor cells were treated with $1 \times 10^5$ of BMDC or BMDM with or without 20 µg/ml AnxA5 protein for 24 h. After co-culture, acquired supernatant were quantitated as mentioned above.

**In vitro recombinant protein functional test**. $2 \times 10^5$ of TC-1 tumor cells were treated with 5ug/ml of cisplatin for 18 h and washed to remove residual cisplatin. The cells were then treated with 0.5 µg/ml of FITC-labeled recombinant AnxA5 or AnxA5-E7 protein, or commercially available FITC-AnxA5 (BD, USA), and analyzed by flow cytometry analysis.

**Analysis of DC maturation and lymph node migration**. To detect migration of CD11c+ dendritic cells into lymph nodes, C57BL/6 mice (5 per group) were injected with $2 \times 10^5$ TC-1 cells/mouse subcutaneously on day 0. Mice were then treated intraperitoneally with 5 mg/kg Cisplatin on days 12 and 15, intravenously with 200jig/mice of Annexin A5 proteins on days 13, 14, 16, and 17, and/or intratumorally with 20jig/mice of fluorescein isothiocyanate (FITC)-labeled E7 peptide on days 16 and 17. PBS was used as control. One day after the last treatment, draining lymph nodes were harvested and homogenized in RPMI-10 using nylon mesh bags. Erythrocytes were lysed and the remaining cells were washed twice with RPMI-10. Cells were stained with APC-labeled anti-CD11c antibody (BD Pharmingen) (1:100 dilution) and analyzed by flow cytometry (Supplementary Fig. 18a).

**Antigen presentation and T cell activation by harvested DCs**. C57BL/6 mice (5 per group) were injected with $2 \times 10^5$ TC-1 cells/mouse subcutaneously on day 0. Mice were then treated intraperitoneally with 5 mg/kg Cisplatin on days 12 and 15 and intravenously with 200jig/mice of AnxA5, 200jig/mice of AnxA5-E7, or 20jig/ mice of E7 long peptide intratumorally on days 13 and 16. On day 18 after tumor inoculation, Inguinal lymph nodes were then harvested from treated mice, and CD11c + cells were sorted from a single-cell suspension of isolated inguinal lymph nodes using CD11c micro-Beads Ultrapure (MACS Miltenyi Biotec, Germany). Isolated CD11c + cells were analyzed by forward and side scatter and gated around a population of cells with size and granular characteristics of dendritic cells. The isolated CD11c+ dendritic cells ($1 \times 10^5$) were incubated with $1 \times 10^6$ CD8+ T cells harvested from the spleens of AnxA5-E7 treated TC-1 tumor-bearing mice for 16 h. The IFN-γ cytokine levels were measured using a mouse Ready-Set-Go ELISA kit (Invitrogen, USA). All results were measured with a plate reader.

**Luciferase-based bioluminescence imaging**. Gaussia luciferase (Gluc) and the substrate coelenterazine (Sigma-Aldrich, Germany) were used to test for Gluc activity in vivo. For the in vivo bioluminescence experiment, mice were injected with $1 \times 10^5$ TC-1 cells. 10 days after tumor challenge, mice were treated with 5jig/ml of cisplatin via intraperitoneal injection. 2 days after cisplatin treatment, 200jig of Gluc or AnxA5-Gluc protein was injected intravenously. 1 day after Gluc or AnxA5-Gluc injection, luciferin substrate was injected intraperitoneally and the bioluminescence of the cells was detected via IVIS Spectrum Imaging System Series 2000. The region of interest from displayed images was designated and quantified as total photon counts using the Living Image 2.50 software (Xenogen).

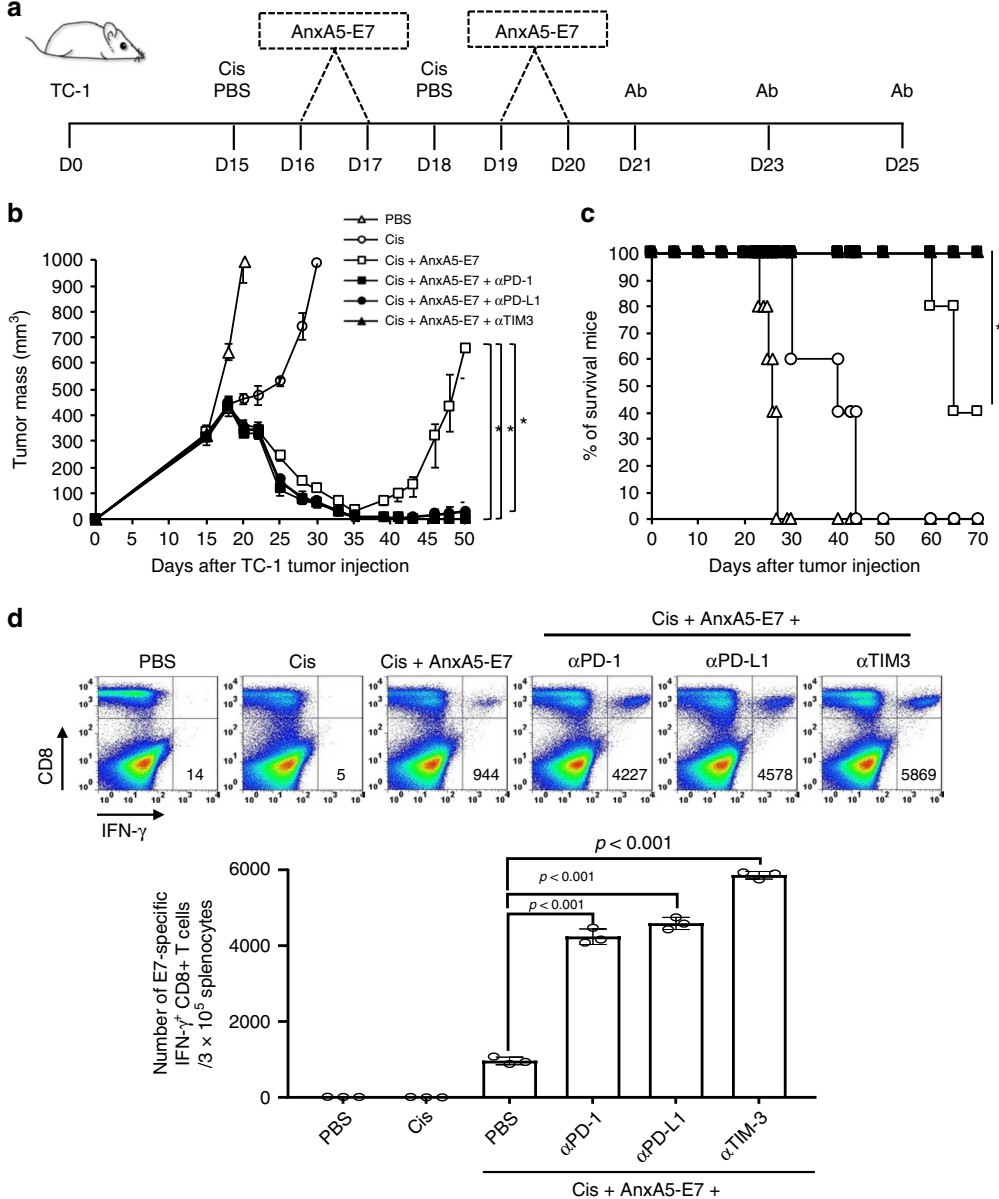

**Fig. 7 Synergy of Annexin A5-E7 fusion protein and immune checkpoint inhibitors.** C57BL/6 mice were injected with $2 \times 10^5$ TC-1 cells/mouse subcutaneously on day 0. Mice were then treated intraperitoneally with 5 mg/kg Cisplatin on days 15 and 18, intravenously with 200jig/mice of AnxA5-E7 proteins on days 16, 17, 19, and 20, and/or intraperitoneally with 200jig/mice of anti-PD-1, anti-PD-L1, or anti-TIM-3 antibodies on days 21, 23, and 25. The treatment groups are as follows: opened triangle - PBS only; opened circle - cisplatin only; opened square - cisplatin and Annexin 5A-E7; closed square – cisplatin, Annexin 5A-E7, and anti-PD-1; closed circle – cisplatin, Annexin 5A-E7, and anti-PD-L1; closed triangle - cisplatin, Annexin A5-E7, and anti-TIM3. **a** Schematic diagram. **b** Line graph depicting TC-1 tumor growth in different treatment groups over time ($n = 10$). P-values were determined by one-way ANOVA and Turkey's test. **c** Kaplan–Meier survival analysis of TC-1 tumor-bearing mice in different treatment groups ($n = 10$), and the overall P-value was calculated by the log-rank test. **d** One week after the last AnxA5-E7 vaccination, spleens of TC-1 tumor-bearing mice in different treatment groups were harvested and analyzed for CD8+IFN-γ+ T cells by flow cytometry analysis. Figure showing representative flow cytometry analysis and bar graph depicting the abundance of CD8+IFN-γ+ T cells in splenocytes of TC-1 tumor-bearing mice in different treatment groups ($n = 3$). P-values were analyzed by Student's t test. The error bars indicate mean ± SD. The results are representative of one of three independent experiments. Source data are provided as a Source Data file.

**In vitro endotoxin-free Annexin A5 experiment**. Recombinant AnxA5 protein (20jig/ml) was boiled 1 h or incubated with proteinase K (100jig/ml) (promega) at 40 °C for three hours or polymyxin B (10jig/ml) (sigma) at room temperature for ten minutes and boiled to eliminate proteinase K effect. Dendritic cells were treated with prepared AnxA5 protein and LPS (100 ng/ml). Cells and supernatants were collected and measured with ELISA (TNF-α and IL-6) respectively.

**Statistical analysis**. All data are expressed as means ± standard deviation (S.D.) and are representative of at least three separate experiments. Results for intracellular cytokine staining with flow cytometry analysis and tumor treatment experiments were evaluated by analysis of variance (one-way ANOVA) and the Tukey–Kramer multiple comparison test. Comparisons between individual data points were made using Student's t tests. Survival distributions for mice in different groups were compared through Kaplan–Meier survival curves, and by use of the log-rank tests. All P values < 0.05 were considered significant. Of note, *, ** and *** indicate P values less than 0.05, 0.01, and 0.001, respectively; N.S., not significant.

**Reporting summary**. Further information on research design is available in the Nature Research Reporting Summary linked to this article.

## Data availability

All data supporting the findings of the study are available within the manuscripts and its Supplementary Information files. All raw data is available from the corresponding author upon reasonable request.

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

## Acknowledgements

The authors thank Dr. T.-C. Wu for his critical review and helpful discussion. This study was supported by the National Research Foundation of Korea (NRF) grant funded by the Korea government (NRF-2016R1A5A2012284 and NRF- 2018R1A2B6008455) and supported by Basic Research Laboratory Program through the National Research Foundation of Korea (NRF) funded by the Ministry of Science, ICT & Future Planning (NRF-2013R1A4A1069575). This study was supported by a grant of Korea Health Technology R&D project through the Korea Health Industry Development Institute (KHIDI), funded by the Ministry of Health & Welfare, Republic of Korea (grant number: HI15C2524). This study was also supported by the National Institutes of Health, National Cancer Institute Specialized Program of Research Excellence (SPORE) in Cervical Cancer grant (NIH/NCI P50CA098252).

## Author contributions

Conception and design: T.H. K., Y.M. P., C.F.H. Development of methodology: T.H.K., J.H.P. Acquisition of data (Animal experiments, in vitro experiments etc.): T.H.K., J.H.P., H.J.P., S.E.L., Y.S.K., G.Y.J. Analysis and interpretation of data (e.g., statistical analysis, biostatistics, computational analysis): T.H.K., J.H.P., A. Y., Y.M.P., C.F.H. Writing, review, and/or revision of the manuscript: T.H.K., J.H.P., A.Y., E.F., B.L., Y.M.P., C.F.H. Administrative, technical, or material support (i.e., reporting or organizing data, constructing databases): T.H.K., J.H.P., H.J.P., S.E.L., Y.S.K., G.Y.J. Study supervision: Y.M.P., C.F.H.

## Competing interests

The authors declare no competing interests.
