## [Peer Review File · Nature Communications]

Reviewers' Comments:

Reviewer #1:

Remarks to the Author:

In their manuscript "Annexin V as an immune checkpoint inhibitor and tumor-homing molecule for cancer treatment" Tae Heung Kang extensively studied the long-known anti-tumor effect of Annexin A5. The convincingly demonstrated that treatment with AnxA5 has a similar potency as canonical immune checkpoint inhibitors in enhancing immune-based anti-tumor therapy.

The experiment are well planned and go far beyond the current state of the art.

I only have problems with the text.

1. The role of PS in non-tumor situations (bacterial and parasite infections) has to be discussed.
2. Annexin A5 is the current name of the molecule under investigation. If it is abbreviated AnxA5 should be used.
3. Ref 4 has to be removed since there are ambiguities and researchers of the clearance community were unable to reproduce the results.
4. There are many grammatical errors.
5. Very extensive revision of the language is required e.g. many sentences are too long and have to be shortened or split to increase readability.
6. Some sentences are rather cryptic and hard to understand, if at all.

Martin Herrmann (FAU Erlangen-Nürnberg; Germany)

Reviewer #2:

Remarks to the Author:

In this manuscript, Kang et al show that systemic administration of Annexin V following chemotherapy can enhance antitumor immune response induced by a tumoral antigen. In agreement with other reports, they show that Annexin V administration could modify the tumor microenvironment inhibiting the immunosuppressive actions of PS exposed in dying tumor cells. They also show that, based in its affinity for PS expressed in chemotherapy-treated tumors, AnnV could be used to target tumor antigens to the tumor site in vivo and improve their immunogenicity eliciting stronger antitumor responses. This vaccination platform can be even improved by its combination with other immune checkpoint inhibitors such as anti-PD1, PD-L1 or anti TIM3 antibodies. This is an interesting and well written work that could support the use of annexin V as an adjuvant or as an anti-tumor therapy based in blockade of the PS checkpoint inhibitor. The results are in agreement with previous literature supporting the use of AnnV as an antitumor agent. The effect of the AnnV in the TME is very suggestive and is one of the values of this work. The potential use of AnnV for in vivo Ag targeting to the tumor is also interesting. However, some aspects should be considered before the acceptance for its publication.

Major point:

One of the main claims of this work is the potential use of AnnV as a "guiding molecule to home vaccine incorporated tumor antigens to the tumor" and improve their immunogenicity. However, no face-to-face comparison with AnnV plus tumor antigen (not linked covalently) has been done. Experiments from figure 5 do not compare the effect of Cys+AnnV-E7 versus Cys +AnnV +E7 LP peptide. Data from figure 4 using the same schedule demonstrate a 100% survival for Cys +AnnV +E7 LP peptide (not linked). Similarly, in Figure 6, no comparison between Dox+AnnV-AH1 versus Dox+AnnV +AH1 peptide (not linked) has been done. This comparison is important to demonstrate that covalent linkage between these two moieties (AnnV and the antigenic peptide) is needed for the activity of AnnV as a guiding molecule and to support author's conclusion. Probably, a comparison using the delayed treatment schedule starting 15 days after tumor challenge could help.

Figure 2. Ann V rescued the PS-mediated immune suppression in TME generated by cisplatin, leading to greater infiltration of CD8 and CD4 T cells into the tumor, and reduced Treg and MDSC.

As it has been described recently using an anti PS antibody (Freimark et al, 2016 CIR), is this effect on Treg and MDSC also observed systemically (i.e. in the spleen)?

In addition to the effect on TNF-alpha or TGF-beta, has AnnV administration any effect on chemokine profile into the TME to explain a lower recruitment of these immunosuppressive cells into the tumor? Also, because its relevance in cellular immune responses, it would be important to see the effect of AnnV on the secretion of IL-12 by BMDC (in Suppl figure 1).

In Figure 2, panel C, Ann V in the absence of Cisplatin has also an important effect on the number of Treg into the TME. What could be the effect of AnnV in the absence of PS expression induced by cisplatin? Has Ann V a direct effect on Tregs as it has been suggested by others (i.e. Nakahashi-Oda et al, Nat Immunol. 2016)?

Supplementary figure 1. It is interesting to see that co-cultures of DC and TC-1 plus AnnV in the absence of Cisplatin induced also a reduction in TGF-beta in BMDC. Is this reduction statistically significant? If so, what would be the mechanism for this reduction?

It is intriguing to see how the guiding of a tumor antigen through AnnV to the surface of the tumor cell (already expressing the tumor antigen) can potentiate the immune response against the antigen. It would be interesting to see the effect of the administration of AnnV on the maturation state of intratumoral or tumor draining lymph node DC. How Ann V affect DCs maturation in vivo (higher expression of molecules such as CD40 or CD56, CD80, CD86 or MHC-II)?

Also it could be interesting to analyze if Ann V linked to antigenic peptides could favor antigen capture by DC in vivo. It could be possible to isolate DC from mice after immunization of Ann V-Ag in tumor bearing mice and see how efficient are presenting the antigen to antigen-specific T cells (i.e. CD8 lymphocytes isolated from tumor bearing mice expressing or not the tumor antigen. (I.e. DC isolated from TC-1 or CT26 tumor bearing mice treated with cisplatin and immunized with Ann V-E7, could be used as APC in vitro in co-cultures with TILs isolated from TC-1 tumors)

The antitumor effect of TGF-beta inhibition in combination with cisplatin and E7 long peptide is very remarkable (Figure 3). The authors propose the inhibition of TGF-beta as a possible mechanism of action of AnnV. It could be of help to study if PS activates Smad3 on T cells and if Ann V is able to inhibit this TGF-b signaling pathway.

Minor points:

Ann V has been produced in E-coli and although potential endotoxin contamination has been checked by LAL, a control using heated AnnV or protease-digested Ann V in some of the in vitro assays could be done to discard activities of "non-proteinaceous" contaminants.

In figure 3D, Cisplatin plus E7 LP plus anti-TGFbeta, induced around 4200 E7 specific IFN-gamma +ve CD8 T cells per 3×10^5 splenocytes. However, in Figure 4D, this experimental group induced only 130 E7 specific IFN-gamma +ve CD8 T cells in the spleen). How variable is this finding. Are these results representative of various experiments?

Reviewer #3:

Remarks to the Author:

I think the authors have done a good job in showing the potential effect of Annexin V protein. My main problem is while reading I have the feeling the authors have build a hypothesis but fail to really interpret the data in a more open way. As an example the authors claim in figure 2B that Annexin reverts the M2 shift induced by cisplating but from the bar graphs this is not clear to me and also no significance levels are mentioned in the graph. Is there a significant increase in the ratio and is this reversed? Also the absolute numbers are still much higher so what is the effect of that? There are more of these kind of interpretation where in my view they interpret the data too much according to what they expected.

That is a pity as the data are quite interesting and worth publishing.

The results should be really restricted to the model used and this needs more mentioning in the paper. They now overestimate the results that this is applicable to more solid tumors whilst their

model I think the limitations section needs more extension whilst the introduction can be shortened

Figures are not well readable for instance figure 2b the legend.

22 October 2019

Dear Reviewers and Editor,

My coauthors and I appreciate the relevant comments that have been raised in response to our manuscript, “Annexin A5 as an immune checkpoint inhibitor and tumor-homing molecule for cancer treatment.” Your assistance has helped us improve the quality of our manuscript. We have fully addressed each reviewers’ questions/comments and amended the manuscript content accordingly. We have also provided point-by-point responses to the critiques raised by the reviewers. All the reviewer questions are in regular text while our responses are in *italics*. In addition, the changes in the revised manuscript are underlined.

Once again, we thank you for your time, consideration, and invaluable guidance.

Sincerely,

Chien-Fu Hung, Ph.D.
Associate Professor
Departments of Pathology and Oncology
The Johns Hopkins University
Tel: (410)-614-4906/Fax: (443)-287-4295
Email: chung2@jhmi.edu

Editor

We feel that your data should be conformed in at least one additional mouse model following up on reviewer#3 comments to prove your data are not restricted to the specific mouse model used in the study.

*We thank the editor for the critical comment. We have performed additional experiments to characterize the effects of Annexin V treatment following chemotherapy in CT26 tumor model. As shown in **revised Figure 6**, cisplatin + Annexin A5 + AH5 peptide (IV) treatment results in significantly better CT26 tumor control, prolonged survival, and increased AH5-specific CD8+ T cells in both the spleen and in the TME of CT26 tumor bearing mice, as compared to cisplatin + AH5 peptide (IV) treatment. Furthermore, cisplatin + Annexin A5-AH5 fusion protein treatment resulted in even better treatment results than that of cisplatin + Annexin A5 + AH5 peptide (IV) treatment.*

*We then characterized the changes in the TME of CT26 tumor bearing mice following the various treatment similar to that for the TC-1 tumor model shown in Figure 2. As shown in the **new Supplemental Figure 13**, CT26 tumor bearing mice treated with cisplatin + Annexin A5-AH5 have significantly less M2 macrophages, Tregs, and MDSCs, as well as more M1 macrophages, CD8+ T cells, and CD4+ T cells in the TME as compared to those treated with cisplatin alone. Similar to that in TC-1 tumor model, we observed an elevated level of TNF- α and a lower level of TGF- β in the tumor of CT26 bearing mice treated with cisplatin + Annexin A5, cisplatin + Annexin A5 + AH5 (IV), or cisplatin + Annexin A5-AH5, as compared to those treated with cisplatin alone. No differences between the levels of IL-10 in the TME or the level of TNF- α , IL-10, or TGF- β in the serum were observed amongst the different treatment groups in the CT26 tumor model.*

*Our data demonstrated that our initial findings in the TC-1 tumor model is consistent in at least one other tumor model, and is not a phenomenon restricted to the TC-1 tumor model alone. We have included these additional findings in the **revised Results section**.*

Reviewer #1

In their manuscript “Annexin V as an immune checkpoint inhibitor and tumor-homing molecule for cancer treatment” Tae Heung Kang extensively studied the long-known anti-tumor effect of Annexin A5. The convincingly demonstrated that treatment with Annexin A5 has a similar potency as canonical immune checkpoint inhibitors in enhancing immune-based anti-tumor therapy.

The experiment are well planned and go far beyond the current state of the art.

I only have problems with the text.

1. The role of PS in non-tumor situations (bacterial and parasite infections) has to be discussed.

We thank the reviewer for this insightful comment. Indeed, the association between PS expression and immunosuppressive apoptotic clearance has been shown to have critical roles not only in the shaping of an immunosuppressive tumor microenvironment, but also in the context of autoimmune disorder development and during pathogen infections.

The natural, intended role of PS externalization by apoptotic cells is to facilitate the effective clearance of apoptotic cell debris while inducing the production of immunosuppressive cytokines to prevent the induction of unwanted inflammatory responses during the homeostatic erythrocytosis of aged, decaying cells. As such, defects in the PS signaling pathway has been

associated with undesired activation of inflammatory and immune responses. Indeed, previous studies have shown that loss of PS – PS receptor signaling in mice led to the emergence of auto-immune diseases resembling systemic lupus erythematosus, and that defects in PS – PS receptor signaling have been identified in some portion of patients with SLE (Kimani SG, et al. *Front Immunol.* 2014).

On the other hand, due to the ability of PS in promoting the “immunosuppressive uptake” by phagocytes, PS has also been utilized by various pathogens including virus, bacteria, and parasites to promote their infectivity while inhibiting the host’s immune system to mount an effective response for pathogen clearance (Birge RB, et al. *Cell Death Differ.* 2016). It has been reported that various viral pathogen engage in a process known as “viral apoptotic mimicry”, in which the virus will display PS on its surface to resemble apoptotic debris to promote its uptake by phagocytes for subsequent infection while inducing the production of immunosuppressive cytokines by the phagocytes to prevent the generation of viral-targeting immune response (Amara A, et al. *Nat Rev Microbiol.* 2015). This “viral apoptotic mimicry” can be achieved by direct incorporation of PS molecules onto the membrane of enveloped viruses (i.e. Ebola virus, dengue virus, vaccinia virus), expression of unique surface molecules that shares structural homologies to PS-receptor binding proteins (i.e. SV40 polyoma virus), or by hijacking PS-containing membrane of the infected cells and exit the cells in a PS-expressing exosome-like manner (i.e. hepatitis A virus). Similarly, it has been reported that various bacteria (i.e. *Bordetella pertussis*, *Treponema pallidum*) and parasites (i.e. *Plasmodium spp*, *Trypanosoma spp*, *Leishmania*) will either directly express PS on its cell surfaces or promote the upregulation of surface PS by the infected and/or bystander cells to inhibit the effective immunologic clearance of infection (Birge RB, et al. *Cell Death Differ.* 2016).

In sum, the regulation of immune response by PS is critically involved in many disease processes including tumor development, auto-immunity, as well as pathogen infections. As such, the appropriate manipulation of PS signaling, such as the use of AnxA5, may serve as a rational approach for the control of these diseases. We have included this discussion in the **revised Discussion section**.

2. Annexin A5 is the current name of the molecule under investigation. If it is abbreviated AnxA5 should be used.

We thank the reviewer for this comment. We have gone through the manuscript and changed the word “AnnexinV” to “Annexin A5”, and the abbreviation “AnnV” to “AnxA5” as suggested.

3. Ref 4 has to be removed since there are ambiguities and researchers of the clearance community were unable to reproduce the results.

We thank the reviewer for this comment. We have replaced the original citation and modified the associated sentence accordingly.

4. There are many grammatical errors. Very extensive revision of the language is required e.g. many sentences are too long and have to be shortened or split to increase readability. Some sentences are rather cryptic and hard to understand, if at all.

We thank the reviewer for this comment. We have substantially edited the manuscript to correct any grammatical errors and to clarify any potentially confusing sentences.

Reviewer #2

In this manuscript, Kang et al show that systemic administration of Annexin V following chemotherapy can enhance antitumor immune response induced by a tumoral antigen. In agreement with other reports, they show that Annexin V administration could modify the tumor microenvironment inhibiting the immunosuppressive actions of PS exposed in dying tumor cells. They also show that, based in its affinity for PS expressed in chemotherapy-treated tumors, AnnV could be used to target tumor antigens to the tumor site in vivo and improve their immunogenicity eliciting stronger antitumor responses. This vaccination platform can be even improved by its combination with other immune checkpoint inhibitors such as anti-PD1, PD-L1 or anti TIM3 antibodies. This is an interesting and well written work that could support the use of annexin V as an adjuvant or as an anti-tumor therapy based in blockade of the PS checkpoint inhibitor. The results are in agreement with previous literature supporting the use of AnnV as an antitumor agent. The effect of the AnnV in the TME is very suggestive and is one of the values of this work. The potential use of AnnV for in vivo Ag targeting to the tumor is also interesting. However, some aspects should be considered before the acceptance for its publication.

Major point:

1. One of the main claims of this work is the potential use of AnnV as a “guiding molecule to home vaccine incorporated tumor antigens to the tumor” and improve their immunogenicity. However, no face-to-face comparison with AnnV plus tumor antigen (not linked covalently) has been done. Experiments from figure 5 do not compare the effect of Cys+AnnV-E7 versus Cys +AnnV +E7 LP peptide. Data from figure 4 using the same schedule demonstrate a 100% survival for Cys +AnnV +E7 LP peptide (not linked). Similarly, in Figure 6, no comparison between Dox+AnnV-AH1 versus Dox+AnnV +AH1 peptide (not linked) has been done. This comparison is important to demonstrate that covalent linkage between these two moieties (AnnV and the antigenic peptide) is needed for the activity of AnnV as a guiding molecule and to support author’s conclusion. Probably, a comparison using the delayed treatment schedule starting 15 days after tumor challenge could help.

*We thank the reviewer for the important suggestion. We have performed additional experiments to compare the antitumor treatment effect of systemic AnxA5-E7 administration vs. systemic AnxA5 + intratumoral E7 long peptide administration, following cisplatin chemotherapy. As shown in the **new Supplemental Figure 10**, while Cis + AnxA5 + E7 LP (IT) results in significantly better tumor control, prolonged mice survival, and generation of E7-specific CD8+ T cells in both the spleen and in TME, Cis + AnxA5-E7 fusion protein results in even better treatment outcomes compared to Cis + AnxA5 + E7 LP(IT).*

*In addition to the TC-1 tumor and E7 antigen model, we also performed experiments to compare the treatment effect of AnxA5-antigenic peptide fusion protein vs. AnxA5 + antigen peptide concomitant administration in CT26 tumor model. Specifically, we treated CT26 tumor bearing BALB/c mice with cisplatin chemotherapy, followed by intravenous AnxA5, AH5 peptide, and/or AnxA5-AH5 fusion protein injection. As shown in the **revised Figure 6**, Cis + AnxA5-AH5 fusion protein treatment results in better CT26 tumor control, prolonged mice survival, as well*

as stronger CD8+ T cell responses, compared to Cis + AnxA5 + AH5 peptide combination treatment.

Together, these new data further support that fusing antigenic peptide to AnxA5 results in an increase in immunogenicity and treatment efficacy compared to co-administration of AnxA5 and antigenic peptide as separate entities. We have included this data in the **revised Results section**.

2. Figure 2. Ann V rescued the PS-mediated immune suppression in TME generated by cisplatin, leading to greater infiltration of CD8 and CD4 T cells into the tumor, and reduced Treg and MDSC. As it has been described recently using an anti PS antibody (Freimark et al, 2016 CIR), is this effect on Treg and MDSC also observed systemically (i.e. in the spleen)?

We performed additional experiment to assess the MDSC and Treg population in the spleen of TC-1 tumor bearing mice treated with cisplatin and/or AnxA5 treatment. Similar to the results observed in the TME, cisplatin chemotherapy results in an increase in the abundance of MDSCs and Tregs in the spleen of TC-1 tumor bearing mice, which are drastically reduced by the administration of AnxA5 (new Supplemental Figure 1). We have included this additional finding in the revised Results Section.

3. In addition to the effect on TNF-alpha or TGF-beta, has AnnV administration any effect on chemokine profile into the TME to explain a lower recruitment of these immunosuppressive cells into the tumor? Also, because its relevance in cellular immune responses, it would be important to see the effect of AnnV on the secretion of IL-12 by BMDC (in Suppl figure 1).

We thank the reviewer for the insightful comment. We have performed additional experiment to assess the level of CCL2 chemokine in the TME of TC-1 tumor bearing mice following cisplatin and/or AnxA5 treatment, since CCL2 was identified as a key mediator for the recruitment of MDSCs into the TME (Qian BZ, et al. Nature. 2011; Chang AL, et al. Cancer Res. 2016). We observed a significant increase in the amount of CCL2 chemokine in the TME of mice that only received cisplatin chemotherapy, whereas the abundance of CCL2 chemokine in the TME was significantly lower in mice treated with cisplatin and AnxA5 (new Supplemental Figure 4), which correlated well with the amount of MDSCs and Tregs observed in the TME of treated mice (Figure 2). As such, we believe AnxA5 treatment following chemotherapy may influence the production of CCL2 chemokine in the TME, thereby reducing the recruitment of immune suppressive cells into the TME.

In addition, we also assessed the secretion of IL-12 by BMDC and BMDM in vitro upon co-culturing with cisplatin treated TC-1 tumor cells and/or AnxA5 protein. As shown in the revised Supplemental Figure 3, additional treatment with AnxA5 significantly increased the secretion of IL-12 by both BMDC and BMDM co-cultured with cisplatin treated TC-1 tumor cells compared to those without AnxA5 treatment.

We have included the additional data into the revised Results Section.

4. In Figure 2, panel C, Ann V in the absence of Cisplatin has also an important effect on the number of Treg into the TME. What could be the effect of AnnV in the absence of PS expression induced by cisplatin? Has Ann V a direct effect on Tregs as it has been suggested by others (i.e. Nakahashi-Oda et al, Nat Immunol. 2016)?

*This is an interesting point. Most tumor cells intrinsically express an elevated level of phosphatidyl serine on its surface as compared to normal cells, even in the absence of cisplatin chemotherapy. Therefore, it is possible that AnxA5 treatment without cisplatin chemotherapy can still lead to some degree of PS signaling blockage, thereby augmenting the immune TME. However, such change in the abundance of Tregs in the TME of tumor bearing mice that are untreated vs. treated with AnxA5 in the absence of cisplatin was not observed in the CT26 tumor model (new **Supplemental Figure 13C**), suggesting that the result observed in the TC-1 tumor model may be due to undetermined variations between the tumor models, or simply and isolated event as no differences in other immune parameters, such as the abundance of MDSCs, macrophages, CD4+ or CD8+ T cells, or various cytokines, were observed in the TME of TC-1 tumor bearing mice that are not treated vs. treated with AnxA5 alone (Figure 2). While we believe that AnxA5 treatment in the absence of chemotherapy can still result in some inhibition of the intrinsic PS signaling exerted by tumor cells, the level of PS expression by tumor cells without chemotherapy may not be high enough to result in differences that are statistically significant.*

*In their study, Nakahashi-Oda et al reported that the interaction between apoptotic epithelial cells and a subset of dendritic cells results in the inhibition of Treg cell proliferation via the binding between PS expressed on the apoptotic epithelial cells to the negative receptor of PS, CD300a, expressed on the DC subset. This finding that the interaction of apoptotic epithelial cells with DCs results in the promotion of inflammatory responses is contrasting to the classical finding in which the binding between PS-expressing apoptotic cells with phagocytes typically results in the induction of immune tolerance, and is likely contributed to the unique properties of intestinal epithelial cells (Pott J, et al. Nat Immunol. 2016). Indeed, Lankry D et al has previously explored the role of PS-CD300a interaction in the context of tumor, and showed that the PS expressed on the tumor cells can bind to CD300a expressed on the NK cells, resulting in the subsequent inhibition of tumor cell killing by NK cells (Lankry D, et al. Eur J Immunol. 2013). In addition, it has been reported that PS can bind to a variety of surface molecules belonging to the TIM gene family (Freeman GJ, et al. Immunol Rev. 2010). Among them, TIM-3 has been shown to be expressed by a certain population of Tregs that are highly efficient suppressor of effector T cells (Gautron AS, et al. Eur J Immunol. 2014). As such, it is possible that AnxA5 has directly manipulating effect on Treg functions by preventing the interaction between PS-expressing apoptotic tumor cells and TIM-3-expressing Tregs. While the detailed characterization of such T cell signaling pathways may be beyond the scope of the current study, we have included this information in the **revised Discussion section**.*

5. Supplementary figure 1. It is interesting to see that co-cultures of DC and TC-1 plus AnnV in the absence of Cisplatin induced also a reduction in TGF-beta in BMDC. Is this reduction statistically significant? If so, what would be the mechanism for this reduction?

As mentioned above, tumor cells intrinsically expressing an elevated level of phosphatidyl serine on its surface as compared to normal cells, even in the absence of cisplatin chemotherapy. Therefore, it is possible that AnxA5 treatment without cisplatin chemotherapy can still lead to some degree of PS signaling blockage, thus reducing the secretion of TGF-beta by DC upon co-culturing with AnxA5 treated TC-1 cells.

*To confirm whether this change is significant, we have repeated the experiment to coculture BMDC and BMDM with TC-1 cells with or without cisplatin and/or AnxA5 treatment. However, as shown in **revised Supplemental Figure 3**, while there seems to be a slight reduction in the level of TGF-beta when coculturing BMDM or BMDC with AnxA5 treated TC-1 cells compared to those without AnxA5 untreated TC-1 cells, this difference is not statistically significant. We reasoned that in the absence of cisplatin chemotherapy, the expression of surface PS by TC-1, although higher than healthy non-tumor cells, may not be significant enough for the AnxA5 treatment to produce a significant change.*

6. It is intriguing to see how the guiding of a tumor antigen through AnnV to the surface of the tumor cell (already expressing the tumor antigen) can potentiate the immune response against the antigen. It would be interesting to see the effect of the administration of AnnV on the maturation state of intratumoral or tumor draining lymph node DC. How Ann V affect DCs maturation in vivo (higher expression of molecules such as CD40 or CD56, CD80, CD86 or MHC-II)?

*To assess whether AnxA5 treatment affects the maturation of DCs in vivo, we treated TC-1 tumor bearing mice with cisplatin and/or AnxA5 as shown in **new Supplemental Figure 2A**. 1 day after the last AnxA5 administration, we harvested the tumor draining lymph nodes and assess the expression of maturation markers CD40 and CD86 by CD11c+ DCs by flow cytometry analysis, and found that cisplatin + AnxA5 treatment resulted in higher frequencies of CD11c+ DCs with high expression of CD40 or CD86 compared to cisplatin or AnxA5 treatments alone (**new Supplemental Figure 2B**).*

*To assess whether AnxA5 treatment affects the migration of DCs from the tumor to the draining lymph node, we further injected TC-1 tumor bearing, cisplatin and/or AnxA5 treated mice with FITC labeled E7 peptide intratumorally (**new Supplemental Figure 2A**). 1 day after the last treatment, we harvested the tumor draining lymph nodes and analyzed the expression of FITC by DCs via flow cytometry analysis. As shown in **new Supplemental Figure 2C**, a greater abundance of FITC+ DCs were observed in the tumor draining lymph nodes of mice treated with cisplatin + AnxA5, as compared to those treated with AnxA5 only or cisplatin only.*

*Together, these new data suggest that AnxA5 treatment enhances the maturation and lymph node migration of DCs in the TME, contributing to the subsequent generation of stronger immune responses. We have included this information in the **revised Results section**.*

7. Also it could be interesting to analyze if Ann V linked to antigenic peptides could favor antigen capture by DC in vivo. It could be possible to isolate DC from mice after immunization of Ann V-Ag in tumor bearing mice and see how efficient are presenting the antigen to antigen-specific T cells (i.e. CD8 lymphocytes isolated from tumor bearing mice expressing or not the tumor antigen. (I.e. DC isolated from TC-1 or CT26 tumor bearing mice treated with cisplatin and immunized with Ann V-E7, could be used as APC in vitro in co-cultures with TILs isolated from TC-1 tumors)

We thank the reviewer for this interesting suggestion. To assess the antigen presentation ability of DCs following AnxA5-Ag fusion protein treatment, we treated TC-1 tumor bearing mice with cisplatin, AnxA5, E7 long peptide, and/or AnxA5-E7 fusion protein. 2 days after the last treatment, we isolated the DCs in the tumor draining lymph node, and co-cultured the harvested DCs with CD8+ T cells harvested from the spleens of TC-1 tumor bearing, cisplatin

and AnxA5-E7 treated mice. As shown in the **new Supplemental Figure 9**, co-culturing the harvested CD8⁺ T cells with DCs obtained from mice treated with Cis + AnxA5-E7 resulted in significantly higher IFN-gamma secretion by the CD8⁺ T cells as compared to those co-cultured with DCs obtained from mice treated with Cis + AnxA5, Cis + E7 LP, or Cis + AnxA5 + E7 LP. This finding support our hypothesis that fusing antigenic peptide to AnxA5 can increase the uptake of antigen by DCs and, coupled with the finding that AnxA5 treated induces maturation and lymph node migration of tumor DCs, contributes to the generation of potent anti-tumor immune response. We have included this information in the **revised Results section**.

8. The antitumor effect of TGF-beta inhibition in combination with cisplatin and E7 long peptide is very remarkable (Figure 3). The authors propose the inhibition of TGF-beta as a possible mechanism of action of AnnV. It could be of help to study if PS activates Smad3 on T cells and if Ann V is able to inhibit this TGF-b signaling pathway.

*We would like to clarify that we are not trying to claim that PS has direct effect on the TGF-beta signaling pathway, or that AnxA5 can directly inhibit TGF-beta signaling pathway. Rather, we believe that a significant portion of the immunosuppressive effect observed by PS is through its ability to induce elevated production of TGF-beta by phagocytes. AnxA5 treatment, by preventing the production of TGF-beta via PS signaling pathway rather than direct inhibition of TGF-beta signaling, induces a TME that is less immunosuppressive, thus favoring the induction of anti-tumor immune response. The anti-TGF-beta treatment explored in **Figure 3** is to examine whether direct reduction of TGF-beta cytokine level in the TME can reproduce the effects observed with AnxA5 treatment. We have revised the **Result section** to make this point more clear, and apologize for causing any potential confusion to the reviewer.*

Minor points:

1. Ann V has been produced in E-coli and although potential endotoxin contamination has been checked by LAL, a control using heated AnnV or protease-digested Ann V in some of the in vitro assays could be done to discard activities of “non-proteinaceous” contaminants.

*To confirm that the AnxA5 proteins used in the current study are not contaminated with endotoxin during the production and purification process, we co-cultured dendritic cells with the purified AnxA5 or purified AnxA5 that has been boiled, treated with proteinase K, or treated with polymyxinB, and assessed the subsequent TNF-alpha cytokine production by the DCs. LPS was used as positive control. As shown in **new Supplemental Figure 14**, no detectable level of TNF-alpha cytokine production were observed in DCs co-cultured with normal AnxA5, boiled AnxA5, or AnxA5 treated with proteinase K or polymyxin B, further supporting that the AnxA5 protein are not contaminated with endotoxins, and that the effects of AnxA5 treatment observed in the current study are due to potential effects of endotoxin contaminants, but by the properties of AnxA5 protein. We have included this information in the **revised Discussion Section**.*

2. In figure 3D, Cisplatin plus E7 LP plus anti-TGFbeta, induced around 4200 E7 specific IFN-gamma +ve CD8 T cells per 3x10⁵ splenocytes. However, in Figure 4D, this experimental group induced only 130 E7 specific IFN-gamma +ve CD8 T cells in the spleen). How variable is this finding. Are these results representative of various experiments?

We thank the reviewer for raising this question. In figure 3, we assessed the abundance of IFN-gamma+ CD8+ T cells in the spleen of treated tumor bearing mice on day 23, which is 1 week after the last E7 LP injection and 5 days after the last anti-TGF-beta antibody administration. In comparison, in figure 4, we examined the abundance of IFN-gamma+ CD8+ T cells in the spleen of treated tumor bearing mice on day 19, which is only 3 days after last E7 LP injection and 1 day after the last anti-TGF-beta antibody administration. We believe observed differences in the abundance of IFN-gamma+CD8+ T cells observed between figure 3 and figure 4 are due to the difference in the time point in which they are assessed. We apologize for any confusion raised by this difference, have modified the figure legends to make the experimental schedule clearer.

Reviewer #3

1. I think the authors have done a good job in showing the potential effect of Annexin V protein. My main problem is while reading I have the feeling the authors have build a hypothesis but fail to really interpret the data in a more open way. As an example the authors claim in figure 2B that Annexin reverts the M2 shift induced by cisplatin but from the bar graphs this is not clear to me and also no significance levels are mentioned in the graph. Is there a significant increase in the ratio and is this reversed? Also the absolute numbers are still much higher so what is the effect of that? There are more of these kind of interpretation where in my view they interpret the data too much according to what they expected.

That is a pity as the data are quite interesting and worth publishing.

The results should be really restricted to the model used and this needs more mentioning in the paper. They now overestimate the results that this is applicable to more solid tumors whilst their model I think the limitations section needs more extension whilst the introduction can be shortened

*We understand the reviewer's concern. We have **revised Figure 2B** to display the total number of M1 and M2 macrophages in the TME of TC-1 tumor bearing mice treated with cisplatin and/or AnxA5. As shown in the revised figure, there is a statistically significant increase in the number of M1 macrophages, as well as a significant decrease in the number of M2 macrophages, in the TME of cisplatin + AnxA5 treated TC-1 tumor bearing mice compared to that of mice treated with cisplatin only. In addition, in the **new Supplemental Figure 13**, we performed additional experiments to characterize the changes in the TME of CT-26 tumor bearing mice treated with cisplatin, AnxA5, AH5 peptide, and/or AnxA5-AH5 fusion protein. As shown in **new Supplemental Figure 13B**, a significant increase in the abundance of M1 macrophage as well as a decrease in the abundance of M2 macrophage were observed in the TME of CT-26 tumor bearing mice treated with cisplatin + AnxA5 compared to that of mice treated with cisplatin alone. No significant difference in the abundance of total macrophage between various groups were observed. The preferential induction of M2 macrophage polarization by PS has also been discussed by existing literatures (Birge RB, et al. Cell Death Differ. 2016), which further supports our current finding.*

*Our additional experiments in the CT26 tumor model (**revised Figure 6** and **new Supplemental Figure 13**) resulted in similar findings as observed in TC-1 tumor model, in that cisplatin + AnxA5-AH5 treatment in CT26 tumor bearing mice resulted in significantly better tumor control, mice survival, antigen-specific CD8+ T cell response, and a more immune*

*stimulating TME. Our new data demonstrates that the finding of the current study is applicable to at least 2 different tumor models, and is not an isolated phenomenon restricted to TC-1 tumor model alone. We have included these additional data in the **revised Results Section**.*

*Nonetheless, we agree that there could be potential variations to our findings in other tumor models that are not tested in the current study, and have included additional discussion suggesting the need to examine the effects of AnxA5-antigenic peptide fusion proteins in more tumor models in the **revised Discussion Section**.*

2. Figures are not well readable for instance figure 2b the legend.

We have revised Figure 2B to display the number (rather than ratio) of M1 and M2 macrophages in the TME of TC-1 tumor bearing mice treated with various modalities, and edited the figure legends to make them easier to understand. We hope the modifications successfully addressed the reviewer's concern and thank the reviewer for the critical feedback.

Reviewers' Comments:

Reviewer #1:

Remarks to the Author:

I am fine with the Revision.

Dr. Martin Herrmann

Department for Internal Medicine 3

University Hospital Erlangen, Germany

Reviewer #2:

Remarks to the Author:

The authors have carried out additional experiments and answered all questions raised by this reviewer. I consider the article acceptable for publication

Reviewer #3:

Remarks to the Author:

I thank the authors for their efforts

10 January 2020

Dear Editor,

My coauthors and I appreciate all the assistance during the submission process and your willingness to consider our manuscript, “Annexin A5 as an immune checkpoint inhibitor and tumor-homing molecule for cancer treatment,” for publication. We have fully addressed each issue and amended the manuscript content accordingly. A point-by-point responses to the issues are provided below. All the raised issues are in regular text while our responses are in *italics*. In addition, we have provided a revised manuscript with track changes to highlight the amended contents.

Once again, we thank you and all the reviewers for your time, consideration, and invaluable guidance.

Sincerely,

Chien-Fu Hung, Ph.D.
Associate Professor
Departments of Pathology and Oncology
The Johns Hopkins University
Tel: (410)-614-4906/Fax: (443)-287-4295
Email: chung2@jhmi.edu

REVIEWERS' COMMENTS:

Reviewer #1 (Remarks to the Author):

I am fine with the Revision.

Reviewer #2 (Remarks to the Author):

The authors have carried out additional experiments and answered all questions raised by this reviewer. I consider the article acceptable for publication

Reviewer #3 (Remarks to the Author):

I thank the authors for their efforts

We thank the reviewers for all the insightful comments and feedbacks.